

# On what scales can GOSAT flux inversions constrain anomalies in terrestrial ecosystems?

Brendan Byrne[1], Dylan B. A. Jones[1,2], Kimberly Strong[1], Saroja M. Polavarapu[3], Anna B. Harper[4], David F. Baker[5,6], and Shamil Maksyutov[7]

[1]Department of Physics, University of Toronto, Toronto, Ontario, Canada
[2]Joint Institute for Regional Earth System Science and Engineering, University of California, Los Angeles, California, USA
[3]Climate Research Division, Environment and Climate Change Canada, Toronto, Ontario, Canada
[4]College of Engineering, Mathematics, and Physical Sciences, University of Exeter, Exeter, UK
[5]NOAA Earth System Research Laboratory, Global Monitoring Division, Boulder, CO, USA
[6]Cooperative Institute for Research in the Atmosphere, Colorado State University, Ft. Collins, CO, USA
[7]Center for Global Environmental Research, National Institute for Environmental Studies, Tsukuba, Japan

**Correspondence:** Brendan Byrne (bbyrne@physics.utoronto.ca)

**Abstract.** Interannual variations in temperature and precipitation impact the carbon balance of terrestrial ecosystems, leaving an imprint in atmospheric $CO_2$. Quantifying the impact of climate anomalies on the net ecosystem exchange (NEE) of terrestrial ecosystems can provide a constraint to evaluate terrestrial biosphere models against, and may provide an emergent constraint on the response of terrestrial ecosystems to climate change. We investigate the spatial scales over which interannual variability in NEE can be constrained using atmospheric $CO_2$ observations from the Greenhouse Gases Observing Satellite (GOSAT). NEE anomalies are calculated by performing a series of inversion analyses using the GEOS-Chem model to assimilate GOSAT observations. Monthly NEE anomalies are compared to "proxies", variables which are associated with anomalies in the terrestrial carbon cycle, and to upscaled NEE estimates from FLUXCOM. Strong agreement is found in the timing of anomalies in the GOSAT flux inversions with soil temperature and FLUXCOM. Strong correlations are obtained (P < 0.05, $R > R_{\mathrm{NINO3.4}}$) in the tropics on continental and larger scales, and in the northern extratropics on sub-continental scales during the summer ($R^2 \geq 0.49$). These results, in addition to a series of observing system simulation experiments that were conducted, provide evidence that GOSAT flux inversions can isolate anomalies in NEE on continental and larger scales. However, in both the tropics and northern extratropics, the agreement between the inversions and the proxies/FLUXCOM is sensitive to the flux inversion configuration. Our results suggest that regional scales are likely the minimum scales that can be resolved in the tropics using GOSAT observations, but obtaining robust NEE anomaly estimates on these scales may be difficult.



# 1 Introduction

Organisms within terrestrial ecosystems have evolved to fit their climatic environment. Anomalous variations in temperature and and precipitation about the mean climate can have significant impacts on the functioning of these organisms (Berry and Bjorkman, 1980; Gutschick and BassiriRad, 2003; Smith, 2011), which can be reflected in anomalies in the carbon balance of ecosystems. In fact, interannual variability (IAV) in the atmospheric growth rate of $CO_2$ is largely explained by changes in the carbon balance of terrestrial ecosystems in response to climate variability (Keeling et al., 1976a, b; Conway et al., 1994; Keeling et al., 1995; Battle et al., 2000). A number of studies have taken advantage of this fact to estimate anomalies in net ecosystem exchange (NEE). In these studies, data assimilation methods are employed to estimate NEE anomalies consistent with measured variations in atmospheric $CO_2$ (Bousquet et al., 2000; Rödenbeck et al., 2003; Bruhwiler et al., 2011; Peylin et al., 2013; Marcolla et al., 2017; Rödenbeck et al., 2018; Shiga et al., 2018a). These studies have generally used $CO_2$ measurements from the global network of in situ instruments for observational constraints. This network provides by far the longest direct record of atmospheric $CO_2$ measurements, with many sites functioning for decades. However, the spatial distribution of sites is inhomogeneous, with sites most densely located in North America and Europe and comparatively few elsewhere. Therefore, in situ observations from the global observation network are relatively insensitive to $CO_2$ fluxes over much of Asia and in the tropics (Byrne et al., 2017), where IAV is the largest. Recently, space-based observations of atmospheric $CO_2$ have provided expanded observational coverage for atmospheric $CO_2$. One of the satellites, the Greenhouse Gases Observing Satellite (GOSAT), has been providing measurements of atmospheric $CO_2$ since 2009. With multiple years of measurements, it is now possible to investigate IAV in the carbon cycle with GOSAT data.

In this study, we investigate interannual flux anomalies estimated from GOSAT measurements using the "flux inversion" method, wherein surface fluxes are estimated from atmospheric $CO_2$ measurements using a tracer transport model and Bayesian inverse methods. A series of flux inversions using the GEOS-Chem four-dimensional variational (4D-Var) data assimilation system (Henze et al., 2007) are performed with different spatial resolutions, prior fluxes and prior error covariances. We also examine the posterior fluxes from two publicly available flux inversion estimates, the GOSAT Level 4 product (Maksyutov et al., 2013) and CarbonTracker, version CT2016 (Peters et al., 2007, with updates documented at http://carbontracker.noaa.gov), which is a flux inversion that assimilates $CO_2$ observations from the surface network. Posterior anomalies in NEE from the inversions are compared with "proxies", variables that are known to be closely associated with IAV in the carbon cycle. Agreement between the anomalies in the inversions and proxies provides corroborating evidence that the inversions are correctly recovering anomalies in NEE (Deng et al., 2016; Mabuchi et al., 2016; Liu et al., 2017). Three proxies are examined: soil temperature ($T_{soil}$) anomalies from the MERRA-2 reanalysis (Reichle et al., 2011, 2017), the Monthly Self-calibrated Palmer Drought Severity Index (scPDSI) (Dai, 2017), and solar-induced chlorophyll fluorescence (SIF) observed by GOME-2 (Joiner et al., 2016). We also use flux data from FLUXCOM, which provides data-driven NEE anomaly estimates (Tramontana et al., 2016; Jung et al., 2017).

Anomalies in temperature and water availability are closely linked to anomalies in terrestrial ecosystems. On the local scales of FLUXNET sites (Baldocchi et al., 2001), temperature and precipitation have both been shown to be major controls on





NEE (see Baldocchi et al. (2018) for a review). On regional and larger scales, stronger correlations have been found with temperature anomalies than with precipitation anomalies (Wang et al., 2013; Jung et al., 2017), particularly in the tropics. Jung et al. (2017) suggest that this is partially due to sub-regional-scale spatial variability in water availability anomalies that compensate, thereby reducing the influence of these anomalies on larger scales, while temperature anomalies are generally more

spatially coherent. Nevertheless, both temperature and water availability anomalies strongly influence NEE anomalies over a wide range of scales. The largest driver of IAV in the carbon cycle is El Niño-Southern Oscillation (ENSO) variability, which most strongly impacts tropical ecosystems (Bacastow, 1976; Bacastow et al., 1980; Bousquet et al., 2000; Ciais et al., 2013). During the warm phase of ENSO (El Niño) large areas of tropical land become dryer and warmer, leading to a net emission of $CO_2$ from the land to the atmosphere, which amplifies the atmospheric $CO_2$ growth rate. During the cold phase of ENSO

(La Niña), much of the tropical land is cooler and wetter than average, leading to anomalously low $CO_2$ growth rates (Jones and Cox, 2005). In the extratropics, there is also significant variability in the carbon balance of terrestrial ecosystems related to temperature and moisture anomalies (Conway et al., 1994; Bousquet et al., 2000). Wunch et al. (2013) show that the summer minima in column-averaged dry-air mole fraction of $CO_2$ ($X_{CO_2}$) observed at northern midlatitude Total Carbon Column Observing Network (TCCON) sites is correlated with surface temperature, indicating that midlatitude positive temperature

anomalies correspond to reduced uptake by the northern extratropical biosphere during the growing season. Similarly, He et al. (2018) show that GOSAT $X_{CO_2}$ anomalies are correlated with indicators surface environmental parameters (such as temperature and drought index). Many studies have examined extreme heatwaves or droughts in the extratropics, such as the 2003 European heatwave (Ciais et al., 2005) and 2010 Russian heat wave and wildfires (Guerlet et al., 2013; Ishizawa et al., 2016). In these cases, positive temperature anomalies and drought conditions result in a release of $CO_2$ from terrestrial

ecosystems to the atmosphere. Zscheischler et al. (2014) show that relatively few extreme events dominate anomalies in gross primary productivity (GPP), and likely NEE. Due to the large seasonal cycle of temperature, precipitation and insolation in the extratropics, the relationship between anomalies in NEE and the proxies is likely a function of time of year. We focus our study of the northern extratropics to the Northern Hemisphere summer (JJA).

    SIF is the emission of radiation by chlorophyll during photosynthesis and thus provides a measure of GPP (Papageorgiou

and Govindjee, 2007; Frankenberg et al., 2011; Guanter et al., 2012; Yang et al., 2015; Damm et al., 2015; Zhang et al., 2016a, b; Wood et al., 2017). Therefore, reduced GPP is associated with reduced SIF, and vice-versa. The relationship to anomalies in NEE is less direct because GPP and ecosystem respiration anomalies are highly correlated (Baldocchi et al., 2018). Therefore, the extent to which SIF anomalies and NEE anomalies should be correlated is not well understood. One study, Shiga et al. (2018b), shows that SIF can be used to inform the spatiotemporal distribution of NEE over North America.

Upscaled NEE estimates from eddy-covariance measurements at flux towers can be used to generate an observation-based estimate of NEE anomalies at regional to continental scales. Kondo et al. (2015) compared the GOSAT L4 product and empirical eddy flux upscaling and found similar responses to climate anomalies in temperate and boreal regions, while poorer agreement was found in the tropics. Here, we use upscaled NEE estimates from FLUXCOM that are generated using upscaling approaches based on machine learning methods that integrate FLUXNET site level observations of $CO_2$ fluxes, satellite remote





sensing, and meteorological data (Tramontana et al., 2016; Jung et al., 2017). For this study, upscaled fluxes generated using multivariate regression splines (MARS) are used. Similar results were found for other upscaling algorithms.

It is important to acknowledge that none of these proxies (or FLUXCOM) should be expected to be perfectly correlated with the true NEE anomalies. Therefore, when there is disagreement between the inversions and proxies, it unclear whether this should be attributed to the inversion NEE or the proxy. Comparisons of flux inversions with the proxies are most useful for identifying "positive" results for which the assimilation of atmospheric $CO_2$ observations has introduced a strong correlation with the proxies. However, these comparisons are less useful for identifying the limits of the inversions with "negative" results, in which the null hypothesis (no correlation) cannot be rejected.

In addition to comparing our flux inversions with the proxies and the FLUXCOM data, we also compare several terrestrial biosphere models (TBMs) with the proxies and the FLUXCOM data. TBMs simulate GPP and ecosystem respiration, and therefore provide estimates of NEE. TBMs are widely applied to simulate projections of the future carbon cycle, however, different models show large disagreements on the relative importance of different processes driving the uptake (Huntzinger et al., 2017). One of the primary goals of atmospheric flux inversions is to provide better constraints on NEE to evaluate these models. Therefore, it is useful to determine whether the agreement between flux inversions and the proxies is closer than the agreement between TBMs and the proxies.

This paper has three main objectives. The first is to quantify the agreement between GOSAT flux inversions and the flux proxies. This will be useful for identifying the utility of using proxies to corroborate flux inversions results. The second is to determine the spatial scales over which the GOSAT inversions constrain flux anomalies. GOSAT observations are expected to best constrain fluxes on large scales, such as the entire tropics. As scales decrease, finer scale structures in the atmospheric $CO_2$ fields are required to constrain fluxes, the smallest scales at which GOSAT observations provide useful constraints on NEE anomalies is currently unclear. We quantify the ability of GOSAT flux inversions to quantify NEE anomalies over a range of spatial scales by, first, examining the agreement between the inversions and proxies over a range of spatial scales and, second, examine the ability of GOSAT inversions to recover true flux anomalies by performing a series of Observing System Simulation Experiments (OSSEs). Monthly anomalies in the tropics are examined throughout the year while anomalies in the northern extratropics are examined during the summer (June-July-August, JJA). The third objective is to quantify the sensitivity of the results for the first two objectives to the inversion setup. This is investigated with a series of GOSAT flux inversions with different model resolution, prior fluxes, and prior error covariances.

This paper is structured as follows. In Sect. 2, we describe the datasets used, flux inversions performed, and how anomalies are calculated. Sect. 3 presents the results of our analysis. Flux inversion NEE anomalies are compared with the proxies in the tropics and northern extratropics individually. We then present an OSSE to examine the smallest spatial scales for which NEE anomalies can be recovered from GOSAT observations. Sect. 4 discusses the agreement in anomalies between the GOSAT flux inversions and proxies, the scales constrained by GOSAT flux inversions, and the sensitivity of these results to the inversion setup. Finally, conclusions are given in Sect. 5.



## 2 Data and methods

### 2.1 FLUXCOM NEE Data

FLUXCOM products are generated using upscaling approaches based on machine learning methods that integrate FLUXNET
site level observations, satellite remote sensing, and meteorological data (Tramontana et al., 2016; Jung et al., 2017). Explana-
tory variables from remote sensing measurements are averaged to produce a mean seasonal cycle (Tramontana et al., 2016),
such that all NEE IAV is introduced by the driving reanalysis (NCEP CRU). In particular, NEE IAV is driven by air temper-
ature, incoming global radiation combined with the mean seasonal cycle of NDVI, and model-based water availability index.
Jung et al. (2017) generate NEE products using several machine learning methods. We downloaded these products from the
Data Portal of the Max Planck Institute for Biochemistry (https://www.bgc-jena.mpg.de). We find that the different algorithms
generally give similar results, therefore we only present results using the multivariate regression spline (MARS) NEE in this
study.

### 2.2 Proxies

#### 2.2.1 Dai Global Palmer Drought Severity Index

The monthly self-calibrated Palmer Drought Severity Index (scPDSI) (Dai, 2017) provides a measure of drought severity on
a $2.5° \times 2.5°$ grid. The scPDSI is computed using observed monthly surface air temperature and precipitation and provides a
measure of surface aridity anomalies and changes on seasonal to longer time scales (Dai et al., 2004; Dai, 2011). We note that
scPDSI may not be a good proxy of soil moisture content over the high latitudes ($>50°$).

#### 2.2.2 SIF

We use the monthly gridded "SIF daily average" product from the NASA Level 3 GOME-2 version 27 (V27) terrestrial
chlorophyll fluorescence data (NASA-SIF, 2016; Joiner et al., 2013, 2016). SIF anomalies are multiplied by negative one to
change the sign of the anomalies, so that positive correlations will be obtained if negative SIF anomalies are correlated with
positive NEE anomalies (emission of $CO_2$ to the atmosphere).

#### 2.2.3 Soil temperature

For the soil temperature proxy, we use soil temperatures from the MERRA-2 (Reichle et al., 2011, 2017) reanalysis. Specifially,
we use the average soil temperature over levels 1–3 (TSOIL1,TSOIL2,and TSOIL3), which reaches a depth of 0.73 m.

#### 2.2.4 NINO 3.4 index

For the phase of ENSO, we use the sea surface temperature (SST) anomaly in the NINO 3.4 region ($5°$ S–$5°$ N, $120°$ S–$170°$ N)
of the tropical Pacific Ocean. This region has been widely used to diagnose ENSO activity. The SST values are calculated from
the Hadley Centre Sea Ice and Sea Surface Temperature data set (HadISST) dataset. The SST anomalies were downloaded



from the National Oceanic and Atmospheric Administration (NOAA) Earth System Research Laboratory (ESRL) website (https://www.esrl.noaa.gov).

## 2.3 Inversion analyses

### 2.3.1 CarbonTracker

We use optimized NEE from the NOAA's CarbonTracker, version CT2016 (Peters et al., 2007, with updates documented at http://carbontracker.noaa.gov). CT2016 optimizes NEE by assimilating in situ observations of boundary layer atmospheric $CO_2$. It employs the ensemble Kalman filter approach to assimilate $CO_2$ with atmospheric chemical transport simulated by the TM5 offline atmospheric model (Krol et al., 2005). For CT2016, TM5 is driven by ERA-Interim assimilated meteorology from the European Centre for Medium-Range Weather Forecasts (ECMWF), with a horizontal resolution of $3° \times 2°$ globally and

$1° \times 1°$ in a nested grid over North America. CT2016 also has IAV in biomass burning. Therefore, when analyzing posterior IAV in CT2016 we examine the IAV in NEE alone (referred to as CT2016) and IAV due to NEE and biomass burning combined (referred to as $CT2016_{w/BB}$).

### 2.3.2 GOSAT level 4 data

We use the GOSAT level 4 data product (Maksyutov et al., 2013) produced by the National Institute for Environmental Studies

(NIES). This product is produced by assimilating NIES Level 2 retrievals of $XCO_2$ into the NIES global atmospheric tracer transport model (NIES-TM) to optimize monthly $CO_2$ fluxes for 64 sub-continental regions. The Vegetation Integrative SImulator for Trace gases (VISIT), a prognostic biosphere model (Ito, 2010; Saito et al., 2014), is used to generate prior biospheric fluxes for the inversion analyses. The GOSAT L4 product also has IAV in biomass burning. Therefore, when analyzing posterior IAV, we examine IAV in NEE alone (referred to as GOSAT L4) and IAV due to NEE and biomass burning combined

(referred to as $GOSAT L4_{w/BB}$).

### 2.3.3 GEOS-Chem

We perform a series of flux inversions using the GEOS-Chem 4D-Var assimilation system (Henze et al., 2007). The GEOS-Chem forward model (www.geos-chem.org) is a global 3-D chemical transport model driven by assimilated meteorology from the Goddard Earth Observing System (GEOS-5) of the NASA Global Modeling an Assimilation Office (GMAO). The native

resolution of the model is $0.5° \times 0.67°$ with 72 vertical levels from the surface to 0.01 hPa, but we run the model at lower resolution (either $2° \times 2.5°$ or $4° \times 5°$, depending on the inversion) with 47 vertical layers. Our model configuration is based on the configuration of Nassar et al. (2011). To optimize surface fluxes, the 4D-Var cost function is minimized as described in Deng et al. (2014) to retrieve monthly scaling factors for prior ocean and terrestrial biosphere fluxes in each grid cell. We use an assimilation window of nine months and keep posterior fluxes from the first six months, then shift the inversion widow

forward six months. Using this method, we optimize NEE spanning 2010–2013 (in addition to a six month spin up inversion starting in July 2009). Monthly ocean fluxes are from Takahashi et al. (2009), anthropogenic emissions are from Andres et al.





(2016), and biomass burning emissions are from the Global Fire Emission Database GFEDv3 (van der Werf et al., 2006). We repeat the 2010 GFEDv3 biomass burning emissions for all years so that there is no prior NEE IAV. Error covariance matrices are taken to be diagonal, such that there are no spatial or temporal covariances. Prior errors are assigned as a percentage of the prior flux estimate rather than an absolute value. We assign 16% error to fossil fuels and 38% error to biomass burning

following Deng et al. (2014).

The GEOS-Chem flux inversions performed in this study are shown in Table 1. The flux inversions are performed with different model configurations to examine the sensitivity of the results to the inversion setup. We perform inversions at two spatial resolutions, $2° \times 2.5°$ and $4° \times 5°$. The spatial resolution is varied to examine whether changes in model transport significantly impact our results. It has previously been shown that there are significant differences in tracer transport as model resolution is

decreased in GEOS-Chem (Yu et al., 2018; Stanevich, 2018). In particular, Stanevich (2018) show that resolution-induced biases of up to 30% can be introduced on the scale of TransCom regions for $4° \times 5°$ relative to $2° \times 2.5°$ for atmospheric methane ($CH_4$) inversions.

The prior error statistics are varied between inversions. The prior error covariance provides a metric of the uncertainty in the prior fluxes. If prior fluxes are well known then small errors are applied. If they are poorly known then large prior

errors are applied and the observations will have a larger impact on the posterior fluxes. However, in general, atmospheric $CO_2$ observations underconstrain the fluxes and additional regularization considerations are required. To prevent overfitting of assimilated observations, prior flux errors are typically tighter than the true uncertainty in NEE fluxes. Therefore, the motivation for varying prior errors in this study is to examine the sensitivity of the posterior flux anomalies to these prior constraints.

Finally, the prior NEE fluxes are varied between flux inversions. For all GEOS-Chem inversions, the prior NEE fluxes

are based on the posterior fluxes from CT2016. CT2016 fluxes are used because they are informed by atmospheric $CO_2$ observations, and thus provide a seasonal cycle of NEE which is closer to observed atmospheric $CO_2$ than a TBM forward run (Byrne et al., 2018). Using prior fluxes which are closer to the observed atmospheric $CO_2$ then justifies tighter prior flux error covariances. We use two different setups of the CT2016 posterior fluxes in the inversions. For four inversions we remove the NEE IAV from the CT2016 fluxes. To do this, the fluxes are averaged over the four years (2010–2013) to generate a mean

seasonal cycle. We then repeat this climatology of NEE fluxes for each year of the inversion. The reason for removing prior NEE IAV is so that all posterior NEE anomalies will be introduced through the assimilation of GOSAT observations. This setup is different from many previous flux inversion studies which have included NEE IAV in the prior fluxes. Therefore, to examine the sensitivity of the posterior NEE IAV to prior NEE IAV, we also perform two inversions that employ three-hourly CT2016 NEE fluxes over 2010–2013 unchanged from those available at http://carbontracker.noaa.gov, other than spatial interpolation to

fit our grid, so that NEE IAV is present on the prior NEE for these inversions. The inversions are given names with a subscript following the convention "model resolution – percentage error applied to prior fluxes – presence of prior NEE IAV", such that, an inversion analysis at $4° \times 5°$ resolution with 100% uncertainty applied to prior fluxes and with prior NEE IAV is named "$GC_{4 \times 5 - 100\% - IAV}$."

For GOSAT observations, we use version v3.5 of the NASA Atmospheric $CO_2$ Observations from Space (ACOS) GOSAT

lite files from the $CO_2$ Virtual Science Data Environment (https://co2.jpl.nasa.gov/#mission=ACOS). Information on the





ACOS retrieval algorithm is available in O'Dell et al. (2012) and Crisp et al. (2012). We selected all measurements from the TANSO-FTS shortwave infrared (SWIR) channel, including ocean glint, high gain and medium gain nadir, which pass the quality flag requirement and have warn levels less than or equal to 15. We generate "super-obs" from the GOSAT retrievals by aggregating the observations to the grid size of our inversion. We generate error estimates using the method described by Ku-

lawik et al. (2016). The reduction in error with aggregation can be calculated using the expression $error^2 = a^2 + b^2/n$, where $a$ represents systematic errors that do not decrease with averaging, $b$ represents random errors that decrease with averaging, and $n$ represents the number of satellite observations that are averaged (Kulawik et al., 2016). Kulawik et al. (2016) give $a = 0.8$ ppm and $b = 1.6$ ppm as mean Northern Hemisphere geometric (co-located) values for GOSAT, and these are the values that we use.

### 2.3.4 Observing system simulation experiments

Five OSSEs are performed, for which pseudo-data are generated by simulating atmospheric $CO_2$ with GEOS-Chem at $4° \times 5°$ spatial resolution and with year-specific NEE from the Joint UK Land Environment Simulator (JULES). The GEOS-Chem $CO_2$ distribution is sampled according to the GOSAT observational coverage. We generate pseudo $XCO_2$ using the GOSAT averaging kernel weighting and apply random errors to the $XCO_2$ pseudo-obs consistent with the error estimates described in

Sect. 2.3.3. The inversion configuration for three of the OSSEs is identical to $GC_{4\times5-44\%}$, $GC_{4\times5-100\%}$, and $GC_{4\times5-100\%-IAV}$, which use the posterior CT2016 fluxes as their prior NEE (see Table 1). These OSSEs are referred to as $OSSE_{CT2016-44\%}$, $OSSE_{CT2016-100\%}$, and $OSSE_{CT2016-100\%-IAV}$, respectively. Two more OSSEs use the same setup as $GC_{4\times5-44\%}$ and $GC_{4\times5-100\%}$, except that for these we use the 2010-2013 mean NEE fluxes from JULES as the prior fluxes. These two OSSEs are referred to as $OSSE_{JULES-44\%}$ and $OSSE_{JULES-100\%}$.

## 2.4 Terrestrial biosphere models

### 2.4.1 JULES

JULES is a community land surface model that has evolved from the UK Met Office Surface Exchange Scheme. Phenology in JULES affects leaf growth rates and timing of leaf growth/senescence based on temperature alone (Clark et al., 2011). Vegetation cover is predicted based on nine plant functional types that compete for space based on their relative productivity

and height but are excluded from growing on agricultural land, based on a fraction of agriculture in each grid cell (Harper et al., 2018). CRU-NCEP was used as model forcing data.

### 2.4.2 VISIT

VISIT is a prognostic biosphere model (Ito, 2010; Saito et al., 2014) that simulates carbon exchanges between the atmosphere and biosphere and among the carbon pools within terrestrial ecosystems at a daily time step. Modeling of plant $CO_2$

assimilation in VISIT is based on a model of light extinction in the canopy, following the formulation of Monsi and Saeki (1953). Autotrophic respiration is formulated as the sum of growth respiration and maintenance respiration. Growth respira-



tion is simulated as the cost to produce new biomass, while maintenance respiration is represented as a function of ground surface temperature. Heterotrophic respiration is the sum of respiration from two layers, litter and humus, which is regulated by soil temperature and soil moisture at each depth. VISIT is driven by reanalysis/assimilation climate datasets provided by the Japan Meteorological Agency (JMA): the Japan 25-year reanalysis (JRA-25)/JMA Climate Data Assimilation System JCDAS)

(Onogi et al., 2007) for the period 1979–present.

### 2.4.3    Carnegie-Ames-Stanford Approach (CASA) Global Fire Emissions Database (GFED) Carbon Monitoring System (CMS) model

The version of the CASA model used here, referred to as CASA CMS, was modified from Potter et al. (1993) as described in Randerson et al. (1996), van der Werf et al. (2006) and Liu et al. (2014). It is driven by MERRA reanalysis and satellite-

observed Normalized Difference Vegetation Index (NDVI) to track plant phenology. These flux estimates were computed at monthly time steps with 0.5° horizontal resolution. Monthly NEE fluxes were downloaded from the CarbonTracker ftp (ftp://aftp.cmdl.noaa.gov/products/carbontracker/co2/CT2016/fluxes/priors/).

### 2.4.4    CASA GFED 4.1

The version of the CASA model used here, CASA GFED 4.1, was modified from Potter et al. (1993) as described in van der

Werf et al. (2017). It is driven by ECMWF reanalysis and satellite-observed NDVI to track plant phenology. These flux estimates were computed at monthly time steps with 0.25° horizontal resolution. Monthly NEE fluxes were downloaded from the CarbonTracker ftp (ftp://aftp.cmdl.noaa.gov/products/carbontracker/co2/CT2016/fluxes/priors/).

### 2.5    Anomalies and correlations

Monthly anomalies are calculated by subtracting the mean 2010-2013 value for a given month from the monthly value for a

specific year. For example, the NEE anomaly for a given month and year is calculated using:

$$ANOM[year, month] = NEE[year, month] - \frac{1}{4} \sum_{i=2010}^{2013} NEE[i, month]. \tag{1}$$

Anomalies are calculated over a range of spatial scales. In each case, the quantity of interest is first averaged into a spatial mean for each month, then anomalies are calculated. The same procedure is followed for JJA anomalies except that the anomaly is calculated over the entire three month period instead of for a single month.

In the tropics, temporal correlations are performed to quantify agreement between NEE anomalies and proxy/FLUXCOM anomalies. We want to test the hypothesis that the assimilation of $CO_2$ observations will significantly increase the correlation between the posterior NEE IAV and the proxies relative to the prior NEE IAV and the proxies. We choose a null hypothesis in which the correlation is zero. This is the correct null hypothesis for flux inversions for which the prior NEE fluxes have no IAV. In flux inversions for which there is IAV in the prior NEE, the correlation between the proxies and prior NEE IAV should

be used as the null hypothesis. However, this would be a significantly more difficult null hypothesis to test, so for simplicity





we choose a null hypothesis of zero correlation for all cases. This is equivalent to testing whether the posterior NEE IAV is significantly correlated with the proxies, regardless of the prior IAV. The threshold for rejection of the null hypothesis ($\alpha$) is chosen to be 0.05, such that the null hypothesis is rejected if the P-value (P) is less than 0.05. We acknowledge that this $\alpha$ threshold is largely arbitrary but is widely used in the literature (Benjamin et al., 2018; Lakens et al., 2018). Throughout the

manuscript, correlations that satisfy this criterion are called "strong". In most cases a second test is performed, in which we test if the correlation between the flux inversion NEE IAV and the proxy is greater than that between the NINO 3.4 index and the proxy, and conclude that the inversion and proxy only show good agreement if both of these thresholds are met. The coefficient of correlation is referred to as $R$.

We also perform a series of linear regressions. In the tropics, linear regressions are performed after aggregating over all trop-

ical land, such that the regression is performed on a single 48 point time series. In the northern extratropics, linear regressions are performed for the set of four JJA anomalies across five sub-continental regions resulting in a 20 point dataset. For all regressions the y-intercept is close to zero, and thus is not reported. The slope of the regressions and coefficient of determination ($R^2$) are reported.

## 3    Results

### 3.1    Tropics

Monthly anomalies in the tropics are examined over a range of spatial scales. The anomalies are aggregated to $4° \times 5°$, $8° \times 10°$, sub-continental regions (shown in Fig. 1), continents, and the entire tropics. Figure 2 shows the mean correlation coefficient between the inversions/TBMs and proxies/FLUXCOM on scales ranging from $4° \times 5°$ grid cells to the entire tropics. Correlations between the NINO 3.4 index and flux proxies are also shown over the range of spatial scales. It is important to consider

correlations between the inversions/TBMs and proxies/FLUXCOM with the influence of ENSO variability in mind, as ENSO is the primary driver of large scale NEE IAV in the tropics. Therefore, to understand how well the flux inversions are capturing NEE IAV it is useful to contrast correlations between the inversion and proxies to correlations between the NINO 3.4 index and proxies.

The correlation between posterior NEE anomalies and proxy/FLUXCOM anomalies increase with aggregation (Fig. 2).

This is expected as atmospheric $CO_2$ observations are expected to best constrain fluxes on large scales, such as the entire tropics. As scales decrease, the signal from variations in the fluxes become weaker and more difficult to constrain with the atmospheric $CO_2$ observations. Correlations between the proxies and the NINO 3.4 index also increase with aggregation. This is because the NINO 3.4 index reflects the large scale ENSO-driven variability in the tropics. Therefore, increasing correlation with aggregation for the NINO 3.4 index is a reflection of the large-scale variability having a larger impact.

To categorize the agreement between the flux inversions and the proxies/FLUXCOM, we state that the flux inversions agrees with a proxy on a given scale only if the correlation is strong (P < 0.05) and greater than the correlation of the proxy/FLUXCOM with the NINO 3.4 index ($R > R_{NINO3.4}$). For the GEOS-Chem inversions, this occurs at regional and larger scales for correlations with FLUXCOM NEE and at continental and larger scales for $T_{soil}$. For the GOSAT L4 inversion,





the correlation only reaches this threshold for $T_{soil}$ at the largest aggregation scale. These results suggest that GOSAT observations provide flux information on continental and larger scales, while regional-scale constraints may be possible. The fact that the correlation coefficient is variable between GOSAT inversions indicates that the agreement between posterior fluxes and the proxies/FLUXCOM is sensitive to the inversion configuration.

We investigate the influence of the inversion configuration by comparing the correlations for the six GEOS-Chem inversion. The $2° \times 2.5°$ inversions generally show slightly better agreement with the proxies/FLUXCOM than the $4° \times 5°$ inversions at regional and continental scales. This could be due to improved transport with higher spatial resolution, however, other aspects of the inversion were changed such as the aggregation of assimilated observations and prior error covariances, which may have also influenced the results. The influence of prior NEE IAV can be evaluated by comparing the $4° \times 5°$ inversions

with and without prior NEE IAV. Correlations are stronger for the inversions without NEE IAV at regional and continental scales. This suggests that the presence prior NEE IAV can degrade posterior NEE IAV and is discussed in more detail in Sec. 4.3.3. The influence of prior error covariances can be evaluated by comparing the inversions with small (44% for $4° \times 5°$ and 66% for $2° \times 2.5°$) and large (100% for $4° \times 5°$ and 200% for $2° \times 2.5°$) prior error. Larger prior errors generally result in larger correlations on regional and larger scales. Large prior errors means that more movement away from the prior during the

inversion is allowed, therefore, better agreement with larger prior errors suggests that the GOSAT data information content is sufficiently large that loose prior errors can be applied without degrading the posterior results by over-fitting the observations.

For CT2016, strong correlations that are greater than those for the NINO 3.4 index are only obtained for SIF and only on the scale of the entire tropics. The reason for the strong correlation with SIF is unclear, but it could be a result of the fact that NEE IAV in one of the CT2016 priors (CASA GFED 4.1) is strongly correlated with SIF. The poorer agreement between CT2016

and the proxies/FLUXCOM than for GOSAT inversions suggests that the network of surface observations does not provide sufficient information to constrain tropical fluxes. However, it is also possible that the inversion setup could play a role.

For the TBMs, correlations are highly model dependent. Of the models, JULES shows the best agreement with the proxies/FLUXCOM. JULES shows strong correlations greater than for the NINO 3.4 index at all scales for FLUXCOM NEE, regionally and over the entire tropics for $T_{soil}$, and regionally for scPDSI. These results suggest that JULES predicts NEE

anomalies in the tropics as well as the GOSAT inversion on continental and larger scales, and may be better at regional and smaller scales. This suggests that it may be challenging to use GOSAT flux inversions to evaluate IAV in JULES NEE. For the other models, less agreement is seen with the proxies/FLUXCOM. The one exception is CASA GFED 4.1 which shows strong correlations with SIF at all scales. This may be due to the fact that this model assimilates greenness indices to estimate GPP fluxes. Anomalies in the greenness indices are likely well correlated with SIF anomalies, therefore, if anomalies in CASA NEE

are driven by anomalies in GPP, it may explain the strong correlation.

We now investigate the magnitude of tropical NEE IAV in the inversions and the TBMs. The magnitude of NEE IAV relative to the proxies/FLUXCOM can be obtained by performing linear regressions of the inversion/TBM NEE anomalies against proxy/FLUXCOM anomalies. Linear regressions are only calculated for the scale of the entire tropics, where the inversions and proxies/FLUXCOM agreed best. Table 2 shows the slope and coefficient of determination ($R^2$) for the regressions. There

is a large amount of variability in the slopes between inversions/TBMs for each proxy/FLUXCOM. The GOSAT inversions





are quite consistent with each other relative to CT2016 and the TBMs. The GOSAT inversions give slopes of 1.03–2.10 for FLUXCOM and 0.061–0.12 for $T_{soil}$. Comparing the GEOS-Chem inversions, the largest differences in the slopes are related to the magnitude of the prior error covariances. Looser prior constraints result in slopes that are 30–80% larger. There are also large differences in the magnitude of posterior NEE IAV between the inversions with and without prior NEE IAV. For example, the slopes for the regression between FLUXCOM and the $4° \times 5°$ GEOS-Chem inversions with prior anomalies are 25–50% larger than for GEOS-Chem inversions without prior NEE IAV. The GOSAT L4 product gives slopes which are consistent with the GEOS-Chem inversions. Furthermore, the agreement between the GOSAT L4 product and proxies (or FLUXCOM) is not sensitive to the inclusion of biomass burning. For CT2016, the best agreement is found with $T_{soil}$ ($0.24 \leq R^2 \leq 0.27$), for which CT2016 gives a smaller slope than the GOSAT inversions. The agreement between CT2016 and proxies/FLUXCOM is not sensitive to the inclusion of biomass burning. For the TBMs, JULES gives good fits with $T_{soil}$ ($R^2 = 0.56$) and FLUXCOM ($R^2 = 0.47$) and gives slopes that are similar in magnitude to the flux inversions. The rest of the TBMs have $R^2$ that are too small to make meaningful comparisons.

### 3.1.1   Detailed analysis of GC$_{2 \times 2.5-200\%}$

We examine the agreement between the GC$_{2 \times 2.5-200\%}$ inversion and the proxies/FLUXCOM in the tropics in more detail. Figure 3 (left column) shows the correlation coefficient for each grid cell between the GC$_{2 \times 2.5-200\%}$ NEE anomalies and the proxy/FLUXCOM anomalies. There are broad positive correlations with the NINO 3.4 index across Central and South America, tropical and southern Africa, and much of the Asia-Pacific. Generally, positive correlations are present between GC$_{2 \times 2.5-200\%}$ and SIF, scPDSI, $T_{soil}$, and FLUXCOM NEE in the Americas, southern Africa, and the Asia-Pacific. Figure 3 (center column) shows the correlation coefficient between the NINO 3.4 index and the proxies over the tropics. Generally, the proxies show strong correlations with the NINO 3.4 index in many of the same regions for which these proxies show strong correlations with GC$_{2 \times 2.5-200\%}$. This suggests that grid-scale correlations between GC$_{2 \times 2.5-200\%}$ and the proxies may be a reflection of the large-scale anomalies across the tropics and do not necessarily imply that the inversion is able to isolate the spatial footprint of ENSO-driven flux anomalies on smaller scales. Alternatively, it is also possible that the proxies themselves do not correlate well with the true NEE at these scales.

We examine whether GC$_{2 \times 2.5-200\%}$ is able to isolate flux anomalies that are separate from the large-scale tropical signal by comparing NEE anomalies for FLUXCOM NEE and GC$_{2 \times 2.5-200\%}$ as a function of time. First, we aggregate GC$_{2 \times 2.5-200\%}$ IAV and FLUXCOM NEE anomalies to the entire tropics and the following continental-scale regions: the Americas, Africa plus the Middle East, and the Asia-Pacific plus the Indian sub-continent (Fig. 1). Figure 4 shows GC$_{2 \times 2.5-200\%}$ and FLUXCOM NEE anomalies as a function of time over the entire tropics and the continental-scale regions. We show raw and smoothed (3-month running mean) monthly NEE anomalies as a function of time. Over the entire tropics, FLUXCOM and GC$_{2 \times 2.5-200\%}$ are highly correlated ($R^2 = 0.69$) (which is shown in Fig. 2). On continental scales, the agreement between FLUXCOM and GC$_{2 \times 2.5-200\%}$ is variable, ranging from $R^2 = 0.08$ for Africa plus the Middle East to $R^2 = 0.61$ for the Americas. All correlations improve after smoothing, suggesting that monthly scale variations are not correctly represented in GC$_{2 \times 2.5-200\%}$, FLUXCOM NEE, or both. We attempt to isolate anomalies specific to each continent by removing the large-scale anomaly





across the entire tropics. This is done by subtracting a mean tropical anomaly (scaled to have the same variance as the continental anomaly) from the continental anomaly using the following equation:

$$\text{DIFF}_{\text{continent}-\text{tropics}} = \text{ANOM}_{\text{continent}} - \text{ANOM}_{\text{tropics}} \times \frac{STD(\text{ANOM}_{\text{continent}})}{STD(\text{ANOM}_{\text{tropics}})}, \qquad (2)$$

where $STD()$ represents standard deviation. $\text{DIFF}_{\text{continent}-\text{tropics}}$ provides an estimate of anomalies in NEE for a given

continent that are not associated with the large-scale ENSO-driven anomalies across the tropics. $\text{DIFF}_{\text{continent}-\text{tropics}}$ is shown
for each continent in Fig. 4e,h,k. The magnitude of the anomalies are reduced after removing the tropical mean anomalies.
Positive correlations are obtained for the Americas ($R^2 = 0.18$), Africa plus the Middle East ($R^2 = 0.07$), and the Asia Pacific
and India ($R^2 = 0.30$). These results suggest that $\text{GC}_{2 \times 2.5-200\%}$ is partially able to isolate NEE anomalies on continental
scales that are separate from the large-scale ENSO-induced variability, and suggests that GOSAT flux inversions can be used

to examine continental scale flux anomalies in the tropics. We note, however, that the the agreement in NEE IAV between
$\text{GC}_{2 \times 2.5-200\%}$ and FLUXCOM is not as strong in Africa and the Middle East.

## 3.2   Northern extratropics

In the northern extratropics, the observational coverage of GOSAT is highly seasonal and so we limit our analysis of anomalies
in the northern extratropics to the summer (JJA), when observational coverage is the best (Liu et al., 2014; Byrne et al., 2017).

Fig. 5 shows the anomalies for the proxies, FLUXCOM NEE, and $\text{GC}_{2 \times 2.5-200\%}$ NEE across the northern hemisphere for
JJA 2010–2013. The proxies and FLUXCOM generally show high coherence in anomalies. Events for which FLUXCOM
NEE gives enhanced emission to the atmosphere also show reduced SIF, increased scPDSI, and increased $T_{soil}$. We have
highlighted (with boxes) major climate anomalies over this time period: the 2010 Russian heat wave, the 2011 drought in
Mexico and southern USA, the 2012 North American drought, and the 2013 California drought. $\text{GC}_{2 \times 2.5-200\%}$ NEE indicates

positive anomalies for all of these major events, suggesting that the inversion can recover sub-continental NEE IAV. However,
there are also instances where the inversion and proxies tend to disagree. For example, in 2010, $\text{GC}_{2 \times 2.5-200\%}$ indicates a
positive anomaly in North America, whereas, the proxies indicate near neutral or negative anomalies.

To examine agreement with the proxies on regional scales, we have aggregated the inversions, the TBMs, proxies, and
FLUXCOM into the five extratropical subcontinental regions shown in Fig. 1. The JJA anomalies in these regions over 2010–

2013 provide 20 data points. We performed a linear regression of these anomalies against the proxies and FLUXCOM. Table 3
shows the slope and $R^2$ values of the regressions. For the GOSAT inversions, the $2° \times 2.5°$ and $4° \times 5°$ with no prior NEE IAV
show the closest agreement with FLUXCOM NEE and $T_{soil}$ ($0.49 \le R^2 \le 0.65$), while the inversions with prior NEE IAV
show substantially poorer agreement ($0.15 \le R^2 \le 0.36$). This is a larger difference between the inversions with and without
prior NEE IAV than was found for the tropics (see Sec. 4.3.3). The inversions with NEE IAV also give a smaller slope indicating

a smaller magnitude of NEE IAV, which is the opposite of what was found in the tropics. Comparing the inversions without
prior NEE IAV, tight prior errors give $0.57 \le R^2 \le 0.65$, whereas loose prior constraints give $0.49 \le R^2 \le 0.62$. As with the
tropics, the inversions with looser prior constraints give larger slopes, suggesting larger NEE IAV.



Comparing the other inversions, the GOSAT L4 product shows agreement with FLUXCOM NEE ($R^2 = 0.33$) and $T_{soil}$ ($R^2 = 0.43$). CT2016 shows poor agreement with all proxies, indicating that this inversion is unable to isolate zonally asymmetric fluxes in the northern extratropics, which is surprising given the high sensitivity of the surface $CO_2$ network to northern extratropical surface fluxes (Byrne et al., 2017). However, consistent with this result, Polavarapu et al. (2018) show that flux in-

versions assimilating measurements from the surface network are less able to recover zonally asymmetric flux signals than flux inversions assimilating GOSAT measurements. CT2016 also includes prior NEE IAV in the inversion, which may negatively impact the posterior NEE IAV, based on the GEOS-Chem inversion results.

For the TBMs, VISIT shows close agreement with FLUXCOM NEE, scPDSI, and $T_{soil}$ anomalies and to a lesser extent SIF anomalies. This is notable as VISIT generally showed poor agreement with the proxies in the tropics. JULES shows close

agreement with $T_{soil}$ anomalies and some agreement with the other proxies. CASA GFED 4.1 shows good agreement with SIF anomalies, but comparatively poorer agreement with the other proxies. CASA GFED CMS shows some agreement with SIF anomalies, but little agreement with the other proxies.

### 3.3 Observing system simulation experiments

We performed a series OSSE experiments to investigate the minimum spatial scales that can be constrained by GOSAT obser-

vations. In these experiments pseudo-observations were assimilated from a GEOS-Chem forward model run which had JULES NEE fluxes prescribed. See Sect. 2.3.4 for additional details of the OSSE setup.

### 3.3.1 Tropics

Figure 6 shows the mean correlation coefficient between the posterior and true NEE anomalies in the tropics over a range of scales. The results are highly reminiscent of the results between the GOSAT inversion and the proxies. The mean correlation

between the posterior and true NEE anomalies increases with aggregation for all OSSEs. Strong correlations are obtained for all OSSEs on regional and larger scales. The inversion setup also has an impact on the correlations between the posterior and true NEE IAV. The largest differences between OSSEs are obtained on regional and continental scales. On these scales, $OSSE_{JULES-100\%}$ has the largest correlation. This suggests that having a climatological seasonal cycle close to the true NEE IAV is important for recovering NEE IAV in the tropics. The inclusion of prior NEE IAV ($OSSE_{CT2016-100\%-IAV}$) does not

appear to significantly degrade the correlation relative to a prior NEE without IAV ($OSSE_{CT2016-100\%}$). In fact, inclusion of prior NEE IAV actually improves the correlations (except for continental-scale), in contrast to what was found with the real data GOSAT inversions. The prior error constraints generally have a large influence on the correlation with the true NEE IAV. Loose prior constraints give better agreement for all OSSEs, consistent with the GOSAT inversions.

On the scale of the entire tropics, we performed linear regressions between the posterior and true anomalies, which are

shown in Table 4. For all regressions, the magnitude of IAV in the posterior fluxes is less than the true NEE IAV (slope of 0.42–0.75). This suggests that the inversions do not recover the full magnitude of NEE IAV. In addition to comparing posterior and true anomalies, we examine the similarities in posterior anomalies between OSSEs. The right column of Table 4 shows the results of linear regressions between posterior and $OSSE_{JULES-100\%}$ NEE anomalies. The OSSEs without prior NEE IAV





show better agreement with $\text{OSSE}_{\text{JULES}-100\%}$ posterior anomalies than the true anomalies. This suggests that the assimilation of pseudo-data is introducing NEE anomalies in a similar way for all OSSEs and recovering the true NEE IAV is primarily limited by the observational coverage rather than the inversion setup. However, differences between the OSSEs and true NEE IAV may also be due to systematic biases introduced due to factors such as uneven observational coverage (Liu et al., 2014; Byrne et al., 2017).

We examine the continental-scale anomalies in detail for $\text{OSSE}_{\text{JULES}-100\%}$, $\text{OSSE}_{\text{CT2016}-100\%}$, and $\text{OSSE}_{\text{CT2016}-100\%-\text{IAV}}$ in Figure 7, which shows the timeseries of continental scale flux anomalies in the tropics for the OSSEs. The correlation between the OSSEs and true anomalies improves after performing a three month running mean, consistent with the GOSAT inversion results. Strong correlations between the OSSEs and true NEE IAV are obtained after removing the mean tropical signal (using equation 2). These results provide further evidence that GOSAT inversions can largely recover continental scale flux anomalies in the tropics.

### 3.3.2 Northern extratropics

Table 4 shows the slope and $R^2$ for linear regressions of flux anomalies from the OSSEs against the true NEE IAV on sub-continental regions in the northern extratropics during JJA. In all cases the slope is less than one, indicating that the OSSEs are not recovering the full magnitude of NEE IAV. The $R^2$ values are less than between the GOSAT inversions and proxies. This may be due to the fact that temporal anomalies in JULES NEE are highly variable month-to-month and may have a shorter temporal correlation length scales than the true anomalies. Comparing the different OSSE setups, the $\text{OSSE}_{\text{CT2016}-100\%-\text{IAV}}$ performs substantially worse than the OSSEs with no prior NEE IAV ($R^2 = 0.15$ versus $R^2 = 0.30$–$0.48$). This is consistent with comparisons between GOSAT inversions and proxies, and suggests that employing prior NEE IAV in the northern extratropics degrades posterior NEE IAV on sub-continental scales during JJA. OSSEs with tighter prior constraints give larger $R^2$, consistent with the GOSAT inversions. OSSEs with JULES mean seasonal cycle also agree better with the true NEE IAV than those which employ the mean seasonal cycle from CT2016.

## 4 Discussion

### 4.1 Implications of correlations between flux inversions and proxies

The results of this study show varying degrees of agreement between anomalies in GOSAT flux inversions and anomalies in proxies and FLUXCOM. We consistently find that $T_{soil}$ and FLUXCOM NEE show the strongest agreement with the flux inversions, whereas scPDSI and SIF show weaker agreement. In this section, we discuss agreement between the proxies and flux inversions in detail.



### 4.1.1 Agreement with $T_{soil}$ and scPDSI

The results show high consistency in the timing of anomalies between $T_{soil}$ and GOSAT flux inversions on continental and larger scales in the tropics, and on sub-continental scales in the northern extratropics during JJA. These results indicate that $T_{soil}$ is a useful proxy for corroborating NEE IAV in flux inversions in both the tropics and northern extratropics. Linear regressions

between GOSAT flux inversion and scPDSI IAV indicate moderate agreement on the scale of the entire tropics ($R^2 \leq 0.27$) and on sub-continental scales in the northern extratropics ($R^2 \leq 0.29$). The GOSAT flux inversion NEE IAV consistently shows closer agreement with $T_{soil}$ anomalies than with scPDSI in both the tropics and northern extratropics. This is consistent with previous research that has mostly shown that NEE IAV is most closely related to temperature anomalies on large scales (Wang et al., 2013; Jung et al., 2017).

Although the results of this study indicate that $T_{soil}$ is a useful metric for corroborating NEE IAV in flux inversions, inferring the sensitivity of NEE anomalies to temperature anomalies directly is not advised for the fits given in Tables 2 and 3. This is because a number of factors have not been considered in this analysis. One factor is that temperature anomalies are also correlated with moisture and biomass burning anomalies. Keppel-Aleks et al. (2014) show that accounting for these covariances results in reduced sensitivity of NEE anomalies to temperature anomalies. A second factor is that the relationship between NEE

anomalies and temperature and moisture anomalies is variable, depending on large scale climate modes. For example, Fang et al. (2017) show that either temperature or precipitation anomalies can be the primary driver NEE anomalies based on ENSO phase. A third factor is that the impact of temperature and moisture on NEE anomalies may be lagged (Braswell et al., 1997). Ecosystems can take a months to years to recover from droughts (Frank et al., 2015; Schwalm et al., 2017; Sippel et al., 2018). Baldocchi et al. (2018) found that flux anomalies at number of FLUXNET sites are negatively correlated with themselves after

a one-year lag, implying a highly oscillatory behavior in the net carbon fluxes from year to year.

This leaves many opportunities for future work to further investigate the relationship between NEE anomalies and climate variability in more detail. A further limit to the comparisons of flux inversions with $T_{soil}$ and scPDSI anomalies in the tropics is that we do not distinguish between seasons. The relationship between NEE, $T_{soil}$ and scPDSI anomalies likely have substantial seasonal differences (Rödenbeck et al., 2018). We encourage future studies to examine the seasonally-dependent relationships

using longer flux inversions, as well as studies which investigate lagged correlations and climate mode relationships between inversion NEE anomalies and temperature and water availability anomalies.

### 4.1.2 Agreement with SIF

It is notable that correlations with SIF are weaker than those with the other proxies. Linear regressions indicate that SIF anomalies show some correspondence to GOSAT flux inversion anomalies on sub-continental scales in the northern extratropics

during JJA ($0.14 \leq R^2 \leq 0.27$), but little agreement is found in the tropics ($R^2 \leq 0.05$). These results are not all that surprising, as it is not clear that one should expect SIF and NEE to be highly correlated, since SIF is a proxy of GPP rather than NEE. It has previously shown than NEE and GPP anomalies are only moderately correlated (Baldocchi et al., 2018). However, we also note that spurious trends have been found in the observations (Zhang et al., 2018), which could impact the calculated anomalies.





Furthermore, due to GOME-2's large field of view, clouds are almost always present for measurements in the tropics. To test if the GOME-2 SIF anomalies used here, we examined the correlation between FLUXCOM MARS GPP and SIF anomalies (Fig. 8). Spatially heterogeneous agreement is found between the two datasets, with the closest agreement occurring over semi-arid regions. However, correlations are generally positive over the majority of the globe, suggesting that IAV from GOME-2

SIF is reliable.

### 4.1.3   Agreement with FLUXCOM NEE

The GEOS-Chem GOSAT flux inversions with no prior NEE IAV showed close agreement with FLUXCOM NEE anomalies in the tropics on regional and larger scales, and in the northern extratropics on regional scales during JJA. This is a remarkable finding as these data-driven estimates of NEE IAV are independent, and agreement between the two estimates provides a strong

indication that the results are robust. Therefore, comparisons with FLUXCOM NEE may provide a method for corroborating results from flux inversion studies. However, it should be noted that the net annual NEE fluxes produced by FLUXCOM are quite unrealistic (Tramontana et al., 2016; Jung et al., 2017), with annual net draw-down by the biosphere of 18–28 $PgC\,yr^{-1}$.

    It may also be possible to evaluate the magnitude of NEE IAV in FLUXCOM NEE through comparisons with flux inversions. Here we compare the magnitude of NEE IAV between the GOSAT flux inversions and FLUXCOM. The slope of the linear

regression between the inversions indicates the relative magnitude of the inversion and FLUXCOM NEE anomalies. Over the entire tropics, the GOSAT inversions give slopes of 1.03–2.10 (mean of 1.56), suggesting that the magnitude of NEE anomalies are underestimated by FLUXCOM NEE. For JJA in the northern extratropics, the GOSAT inversions give slopes of 0.79–1.59 (mean of 1.31), again suggesting that the magnitude of NEE anomalies are underestimated by FLUXCOM. Furthermore, the OSSEs suggested that the inversions do not recover the full magnitude of NEE IAV, providing further evidence that FLUXCOM

underestimates the magnitude of NEE IAV. This result is consistent with previous studies which indicate that FLUXCOM underestimate the magnitude of NEE IAV (Jung et al., 2011, 2017).

### 4.2   Scales constrained

We investigated the agreement between monthly anomalies in flux inversions and proxies/FLUXCOM over a range of spatial scales in the tropics. The results showed that the agreement between the inversions and the proxies/FLUXCOM were

scale-dependent, which was corroborated by OSSEs. Here we synthesize these results and discuss the ability of GOSAT flux inversions to recover IAV in NEE over the range of scales examined in this study.

    The results provide strong evidence that GOSAT flux inversion can constrain monthly flux anomalies on the scale of the entire tropics. All of the GEOS-Chem GOSAT flux inversions obtained $R^2 \geq 0.55$ for linear regressions with $T_{soil}$, and $R^2 \geq 0.51$ with FLUXCOM NEE. The OSSEs provide further evidence that the true NEE anomalies could be recovered, as linear

regressions between the posterior and true anomalies give $R^2 \geq 0.53$. These results provide strong evidence that the GOSAT inversions are recovering the timing of tropical NEE anomalies, however, there is less agreement on the magnitude of flux anomalies over the tropics. The OSSEs indicate that GOSAT flux inversions can recover 42–68% of the magnitude of NEE anomalies, depending on the inversion setup.





On continental scales in the tropics, the results suggest that GOSAT flux inversion can constrain monthly flux anomalies. The GEOS-Chem inversions show good agreement with FLUXCOM NEE and $T_{soil}$ anomalies. However, the agreement between the inversions and proxies/FLUXCOM on this scale is strongly influenced by the large-scale ENSO anomalies. We isolated the continental scale anomalies by subtracting a mean tropical anomaly for $GC_{2\times2.5-200\%}$ and FLUXCOM (Fig. 4), and for

the OSSEs (Fig. 7). We found that the anomalies were still correlated after removing the mean tropical signal, suggesting that the continental-scale anomalies are largely recovered in the inversions. However, we also found that the inversion setup can have a significant influence on posterior anomalies on continental scales. The strongest correlations between the proxies and inversions were obtained with higher resolution, looser prior constraints, and no prior NEE IAV. Similarly, OSSEs showed the best agreement with the true NEE IAV when looser prior constraints were employed, but suggested that the presence

of prior NEE IAV generally improved agreement with the true NEE IAV. The OSSEs also showed that correlations with the true NEE IAV were improved on continental scales when the prior mean seasonal cycle was closer to the true NEE IAV. Overall, these results suggest that GOSAT observations contain information on continental-scale NEE anomalies in the tropics; however, recovering the correct NEE IAV from these observations may be sensitive to the flux inversion setup. Furthermore, the magnitude of NEE anomalies are likely underestimated.

On regional scales in the tropics, the results were more ambiguous. The GOSAT inversions generally showed good agreement with FLUXCOM NEE IAV on regional scales, but only marginal agreement with $T_{soil}$. The OSSEs also indicate marginal ability to recover regional scale fluxes. From these results, we caution against making conclusions about NEE IAV on regional scales in the tropics using GOSAT flux inversions without corroborating evidence. On smaller scales, correlations do not meet the threshold of $P < 0.05$.

In the northern extratropics during JJA, the results of this study suggest that regional-scale constraints are possible. We found that large flux anomalies due to major climate events are recovered in the inversion for $GC_{2\times2.5-200\%}$ (Fig 5), while linear regressions showed close agreement for the GOSAT flux inversions with FLUXCOM NEE and $T_{soil}$. However, we also found evidence that the posterior NEE IAV was sensitive the the inversion setup. The inversion analyses with prior NEE IAV ($GC_{4\times5-44\%-IAV}$, $GC_{4x5-100\%-IAV}$, and GOSAT L4) showed weaker agreement with the proxies relative to the inversions

without prior NEE IAV. Similarly, the OSSEs showed prior NEE IAV reduced agreement between the posterior and the "true" NEE IAV in the northern extratropics during JJA.

## 4.3   Influence of the inversion configuration

### 4.3.1   Model horizontal resolution

The results of this study indicate that the spatial resolution of the model used in the inversion analysis ($2° \times 2.5°$ or $4° \times 5°$)

has a relatively minor impact on posterior NEE anomalies. This somewhat surprising since recent studies (Yu et al., 2018; Stanevich, 2018) have shown significant transport differences for different resolution versions of GEOS-Chem. Also, Deng et al. (2015) showed that there are large biases in $CO_2$ in the upper troposphere and lower stratosphere in GEOS-Chem that impact inferred flux estimates. It is possible that, although the model transport errors influence the flux estimates, the resolution-





dependent transport processes are not sensitive to NEE IAV for the time period considered here. It could also be related to the information content of GOSAT observations. As we have shown in this study, GOSAT observations only constrain NEE IAV on regional and larger scales. If transport errors have the largest impact on smaller scales, it may explain why model resolution did not have a major impact on our results.

### 5  4.3.2  Prior error covariances

All of the GEOS-Chem inversions were performed with tight (44% for $4° \times 5°$ and 66% for $2° \times 2.5°$) and loose (100% for $4° \times 5°$ and 200% for $2° \times 2.5°$) prior error covariances. The prior error covariances generally had a significant impact on the posterior NEE IAV. In the tropics, inversions with loose prior constraints gave larger correlations with $T_{soil}$ and FLUXCOM NEE on regional and continental scales. Similarly, for the OSSEs, looser prior constraints gave larger correlations with the true NEE IAV on regional and continental scales. This suggests that the information content of the GOSAT observations is sufficiently large in the tropics that prior error covariances of 100% for $4° \times 5°$ or 200% for $2° \times 2.5°$ can be applied without degrading the posterior results by over-fitting the observations.

In the northern extratropics, the inversions with tighter prior constraints gave larger correlations with $T_{soil}$ and FLUXCOM NEE on regional and continental scales. Similarly, tight prior constraints gave larger correlations with the true NEE IAV for the OSSEs. These results are the opposite of what was found for the tropics, and suggests that tighter error constraints (as a percentage of NEE) should be applied in the northern extratropics than in the tropics. These results suggest that, when the prior error covariances are loose in the northern extratropics, the inversion over-fits the GOSAT observations which degrades the agreement with proxies (or the true NEE IAV for OSSEs).

The largest impact of varying the prior error covariances is in the magnitude of posterior NEE IAV. When loose prior constraints are applied the magnitude of NEE anomalies increases by 30–80% (15–30% for OSSEs) in the tropics and 5–60% (0–30% for OSSEs) in the northern extratropics. These results imply that care should taken when making conclusions about the magnitude of NEE anomalies from this analysis. Based on the OSSEs, it seems likely that the inversions underestimate the magnitude of NEE IAV on all scales, but the inversions with looser prior constraints result in a more realistic magnitude of NEE IAV. This suggests that there is a trade off between obtaining a more realistic magnitude of IAV using looser constraints and obtaining more realistic timing of anomalies with tighter prior constraints.

### 4.3.3  Prior fluxes

We investigated the influence of prior NEE IAV on posterior NEE anomalies in flux inversions by performing inversions with prior NEE IAV ($GC_{4\times5-100\%-IAV}$ and $GC_{4\times5-44\%-IAV}$) and without prior NEE IAV ($GC_{4\times5-100\%}$ and $GC_{4\times5-44\%}$), as well as OSSEs with and without prior NEE IAV. In the tropics, the impact of prior NEE IAV is generally small. For the GOSAT inversions, the presence of prior NEE IAV degrades agreement with the proxies on all scales. In the OSSEs, the presence of prior NEE IAV degrades the agreement with the true NEE IAV on continental scales, but improves agreement on regional scales and over the entire tropics. In the northern extratropics, the presence of prior NEE IAV has a large negative impact on agreement with proxies for GOSAT inversions and on agreement with the true NEE IAV in the OSSEs.



Why does the presence of prior NEE IAV degrade the posterior NEE IAV for many of these inversions? Presumably, the reason is related to the fact that the observations under-constrain NEE IAV, such that the prior NEE IAV strongly influences the spatiotemporal distribution of IAV in the posterior NEE. To investigate this, we examined how closely the posterior NEE IAV resembles the prior NEE IAV. Figure 9 shows the agreement between the posterior and prior NEE IAV for $GC_{4x5-100\%-IAV}$

in the tropics and northern extratropics. Posterior NEE IAV is strongly correlated with IAV in the prior NEE, particularly on smaller scales. The fact that correlations between the prior and posterior NEE IAV are strong at $4° \times 5°$ and $8° \times 10°$ is not surprising, as the NEE fluxes are strongly under-constrained at these spatial scales. However, the correlation with the prior NEE IAV is substantially larger than with FLUXCOM on regional ($R^2 = 0.55$ versus $R^2 = 0.15$) and continental ($R^2 = 0.46$ versus $R^2 = 0.26$) scales as well. This suggests that NEE IAV is still under-constrained even on continental scales. Only on the scale

of the entire tropics is the correlation with the prior NEE ($R^2 = 0.42$) less than with the proxies ($R^2 = 0.61$ for FLUXCOM NEE and $R^2 = 0.56$ for $T_{soil}$), indicating that the observations are influencing the posterior NEE IAV more than the prior NEE IAV. These results suggests that the impact of prior NEE IAV on the inversion is likely strongly dependent on how well the prior NEE IAV reflects the true NEE IAV. Realistic prior NEE IAV would likely improve the posterior NEE IAV, conversely, unrealistic prior NEE IAV will degrade the posterior NEE IAV. This implies that the realism of the prior NEE IAV should be

investigated before including it in an inversion analysis. If the objective of the experiment is to examine the timing of posterior NEE IAV introduced through the assimilation of observations, then we recommend that annually-repeating prior fluxes be used to investigate NEE IAV. However, a trade off in using annually-repeating prior fluxes is that the magnitude of NEE IAV will likely be significantly underestimated.

We also investigated the impact of the prior mean seasonal cycle on posterior NEE IAV. We performed a series of OSSEs

to examine the impact of the mean seasonal cycle of the prior fluxes on the inversion and found that correlations with the true NEE IAV were significantly improved on continental scales when the mean seasonal cycle was closer to the true NEE IAV. In particular, $OSSE_{CT2016-100\%}$ gives much weaker correlations with the true NEE IAV than $OSSE_{JULES-100\%}$ after removing the mean tropical signal (Fig. 7). These results suggest that it is important to use prior fluxes with a realistic seasonal cycle to recover IAV in NEE from GOSAT observations.

## 25  5   Conclusions

In this study, we examined the constraints on interannual anomalies in NEE provided by GOSAT observations by performing a series of flux inversions. We addressed three main objectives in this analysis. The first objective was to quantify the agreement between GOSAT flux inversions and flux proxies, which are associated with IAV in the terrestrial carbon cycle, and FLUXCOM NEE. We found strong correlations ($P < 0.05$, $R > R_{NINO3.4}$) with FLUXCOM NEE and $T_{soil}$ in the tropics on continental

and larger scales, and in the northern extratropics on sub-continental scales during the summer ($R^2 > 0.49$), when there is no prior NEE IAV. These results demonstrate that both FLUXCOM NEE and $T_{soil}$ can be useful tools for corroborating flux inversion results. We found flux anomalies from GOSAT inversions were less correlated with scPDSI and SIF. For scPDSI we found some agreement on the scale of the entire tropics ($R^2 \leq 0.27$) and on sub-continental scales in the northern extratropics





($R^2 \leq 0.29$). For SIF, there was some agreement on sub-continental scales in the northern extratropics during JJA ($0.14 \leq R^2 \leq 0.27$), however, little agreement was found in the tropics ($R^2 \leq 0.05$).

The second objective was to determine the spatial scales over which the GOSAT inversion constrain flux anomalies. In the tropics, we found that continental and larger scale flux anomalies can be captured in GOSAT inversions. This conclusion is
supported by strong agreement ($P < 0.05$, $R > R_{\mathrm{NINO3.4}}$) with $T_{soil}$ and FLUXCOM NEE, and a series of OSSEs which showed that the true NEE IAV can be largely recovered on these scales. On regional scales in the tropics, the GOSAT flux inversions showed some agreement with the proxies and FLUXCOM, but the OSSEs indicated that the GOSAT observations likely underconstrain NEE IAV on these and smaller scales. In the northern extratropics, we found that flux anomalies are recovered by GOSAT flux inversions on sub-continental regions during JJA. Strong agreement was found with anomalies in
$T_{soil}$ ($0.57 \leq R^2 \leq 0.65$) and FLUXCOM NEE ($0.49 \leq R^2 \leq 0.65$), when no prior NEE IAV is used. OSSEs supported these findings, indicating that GOSAT observations can recover regional scale flux anomalies in the northern extratropics during JJA.

The third objective was to quantify the sensitivity of the results from the first two objectives to the inversion setup. We found that the agreement between the flux inversions and proxies can be sensitive to the inversion setup. Posterior flux anomalies were most sensitive to the prior fluxes and error covariances. In general, the inclusion of prior NEE IAV from CT2016 in the
inversion degraded the agreement with FLUXCOM NEE and $T_{soil}$, particularly in the extratropics. This result was supported by the OSSEs in the northern extratropics but not in the tropics. We compared the impact of the mean seasonal cycle on the posterior NEE IAV by performing OSSEs and found that having a prior climatological seasonal cycle that was close to the true NEE IAV improved posterior NEE anomalies on continental scales in the tropics. The prior error constraints also had a significant impact on the results. We found that looser constraints in the tropics gave better agreement with the proxies, while
tighter constraints in the northern extratropics gave better agreement with the proxies. The magnitude of the prior constraints had a large impact on the magnitude of NEE anomalies. Also, the OSSEs showed that the magnitude of NEE anomalies are underestimated even with loose prior constraints. These results indicate that the prior fluxes and error covariances need to be carefully considered. The inclusion of prior NEE IAV is an important factor to consider. Including prior NEE IAV may produce a more realistic magnitude of NEE IAV in posterior fluxes but could also degrade the correlation between the posterior and
true NEE IAV. If prior NEE IAV are included in future inversions, attempts should be made to test the realism of the prior NEE IAV. If the objective of the experiment is to examine the timing of posterior NEE IAV introduced through the assimilation of observations, then we recommend that annually-repeating prior fluxes be used to investigate NEE IAV. The mean seasonal cycle of the prior NEE fluxes is also an important factor in the inversion, and the realism the seasonal cycle should be evaluated before performing the inversion. The results also indicate that defining the prior error covariance to be a fraction of the prior
flux may produce either overfitting of GOSAT data in the northern extratropics or underfitting of the data in the tropics.

Overall, our results show that $T_{soil}$ and FLUXCOM NEE are useful for evaluating NEE IAV in flux inversions. Furthermore, comparisons with the anomalies in $T_{soil}$ and FLUXCOM NEE suggest that GOSAT flux inversions are useful tools for constraining IAV in NEE on continental and larger scales in the tropics, and on regional scales in the northern extratropics during JJA.



*Competing interests.* No competing interests are present

*Acknowledgements.* Funding for this work was provided by the Canadian Space Agency, NSERC, and Environment and Climate Change Canada. CarbonTracker CT2016 results were provided by NOAA ESRL, Boulder, Colorado, USA from the website at http://carbontracker.noaa.gov. CASA GFED 4.1 and CASA CMS NEE fluxes were also downloaded from the CT2016 website. The GOSAT L4 product and VISIT NEE were downloaded from the GOSAT Data Archive Service (https://data2.gosat.nies.go.jp). Dai Global Palmer Drought Severity Index was downloaded from the Research Data Archive at the National Center for Atmospheric Research, Computational and Information Systems Laboratory (https://doi.org/10.5065/D6QF8R93), accessed 15 Aug 2017. NASA GOME-2 SIF products were obtained from Aura Validation Data Center [http:/avdc.gsfc.nasa.gov]. FLUXCOM products were obtained from the Data Portal of the Max Planck Institute for Biochemistry [https://www.bgc-jena.mpg.de]. MERRA-2 products were downloaded from MDISC [https://disc.sci.gsfc.nasa.gov], managed by the NASA Goddard Earth Sciences (GES) Data and Information Services Center (DISC). The GEOS-Chem forward and adjoint models are freely available to the public. Instructions for downloading and running the models can be found at http://wiki.geos-chem.org/. ACOS GOSAT lite files were obtained from the $CO_2$ Virtual Science Data Environment [https://co2.jpl.nasa.gov/#mission=ACOS]. The SST anomalies were downloaded from the National Oceanic and Atmospheric Administration (NOAA) Earth System Research Laboratory (ESRL) website (https://www.esrl.noaa.gov). We thank Martin Jung, Andy Jacobson, Joanna Joiner, Randy Kawa, and Jim Collatz for helpful comments on this manuscript.



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



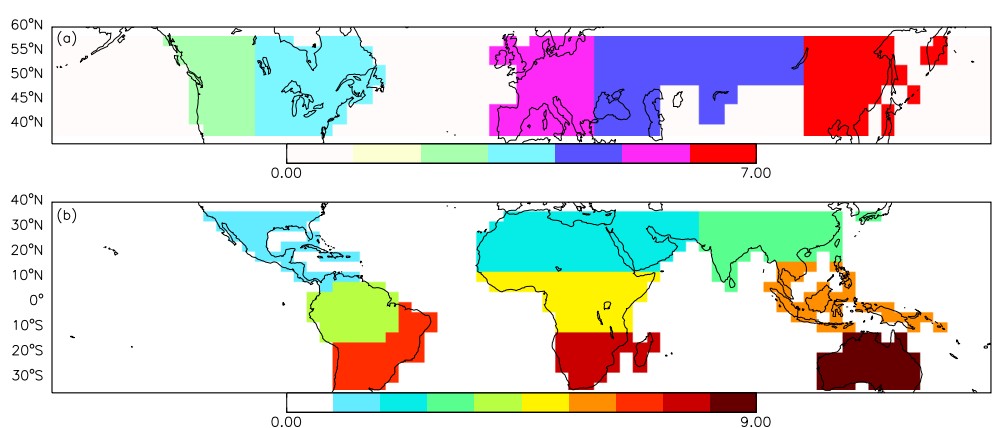

**Figure 1.** Land area at $4° \times 5°$ resolution grouped into sub-continental regions in (a) the northern extratropics and (b) the tropics. In the tropics, we generate three continents by combining the regions in the Americas, Africa and the Middle East, and the Asia-Pacific and Indian sub-continent.





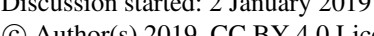

**Figure 2.** Correlation in the tropics over a range of scales for models and inversions with (top) NINO 3.4 index, (second) $(-1) \times$ SIF, (third) scPDSI, (fourth) $T_{soil}$, and (bottom) FLUXCOM NEE in the tropics. Squares represent correlations with terrestrial ecosystem model NEE IAV: VISIT (cyan), JULES (blue), CASA GFED CMS (green), CASA GFED 4.1 (magenta) and the black circle shows the mean correlation of the models. Triangles represent correlations with the GOSAT flux inversion NEE IAV: GOSAT L4 (cyan up-triangle), $GC_{4 \times 5-44\%-IAV}$ NEE IAV (green up-triangle), $GC_{4 \times 5-100\%-IAV}$ NEE IAV (green down-triangle), $GC_{4 \times 5-44\%}$ NEE IAV (red up-triangle), $GC_{4 \times 5-100\%}$ NEE IAV (red down-triangle), $GC_{2 \times 2.5-66\%}$ NEE IAV (orange up-triangle), and $GC_{2 \times 2.5-200\%}$ NEE IAV (orange down-triangle). The green star show the correlation with CT2016 NEE IAV. The grey circle shows the correlation with the NINO 3.4 index. Dashed black lines indicate the correlation required for an $\alpha$ of 0.05, therefore, all correlations greater than the dashed black line indicate P<0.05.





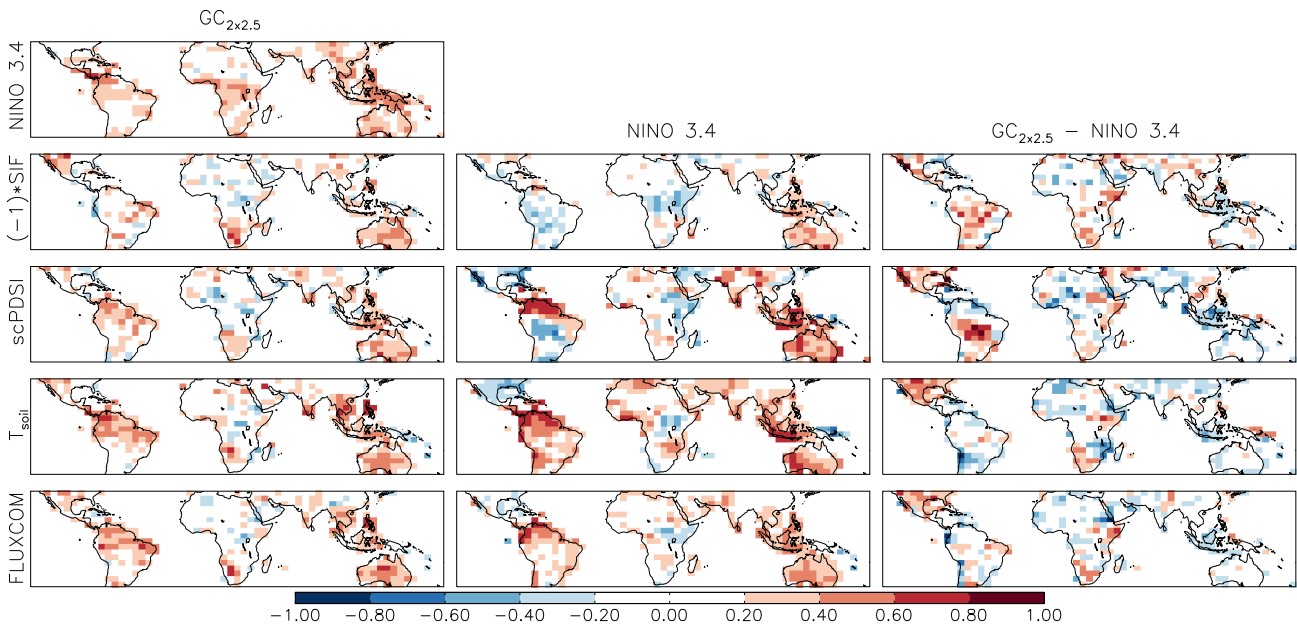

**Figure 3.** Correlations of monthly anomalies over tropical land at $4° \times 5°$ spatial resolution. Columns show coefficient of correlation ($R$) of (left) $GC_{2 \times 2.5 - 200\%}$ NEE IAV, (center) NINO 3.4 index, and (right) the difference between the two with (top row) the NINO 3.4 index, (second row) $(-1) \times$SIF, (third row) scPDSI, (fourth row) $T_{soil}$, and (bottom row) FLUXCOM NEE.





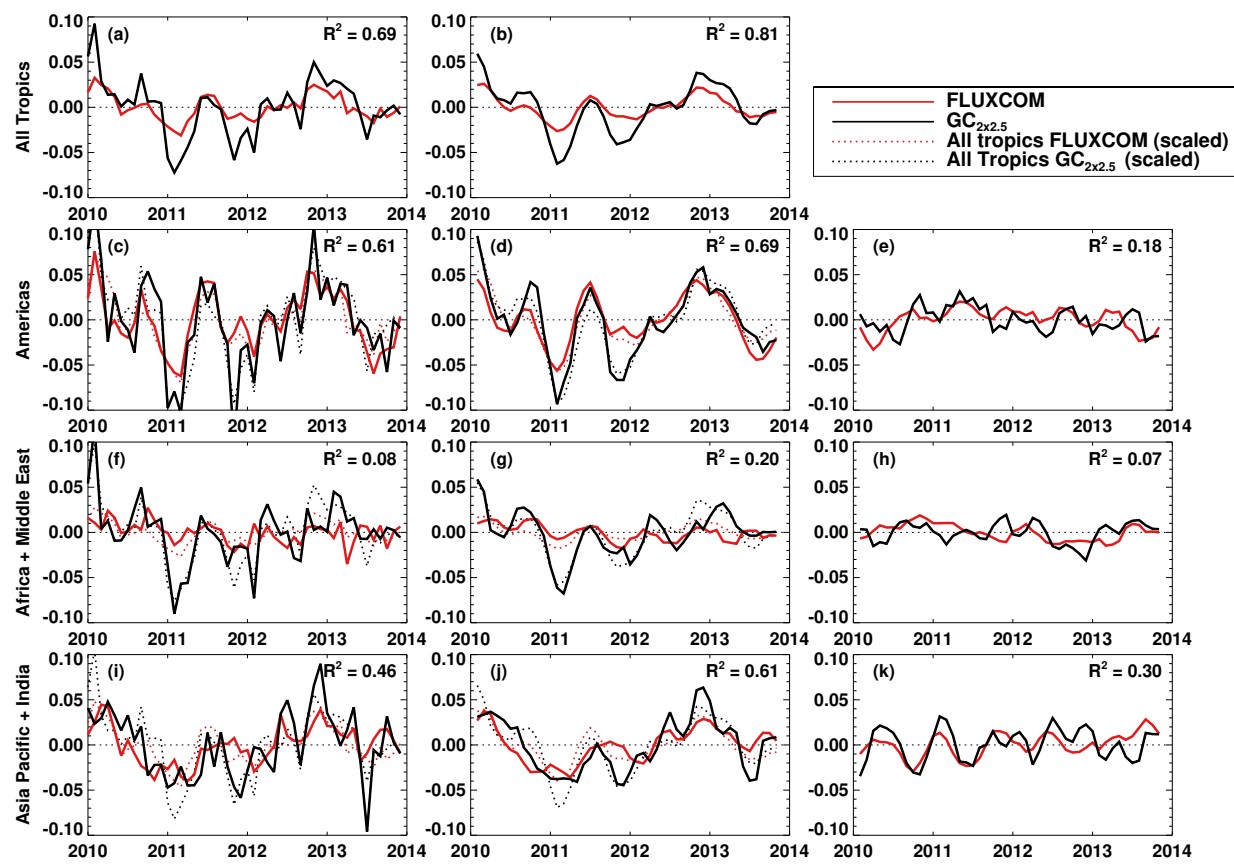

**Figure 4.** NEE anomalies (gC m$^{-2}$ day$^{-1}$) for FLUXCOM and GC$_{2\times2.5-200\%}$ in the tropics. (left column) Monthly anomalies, (center column) smoothed (3-month running mean) monthly anomalies, and (right column) DIFF$_{\mathrm{continent-tropics}}$ (see Sect. 3.1.1 to see how this is calculated) for (a–b) the entire tropics, (c–e) the Americas, (f–h) Africa and the Middle East, and (i–k) the Asia Pacific and Indian sub-continent. For each sub-plot, $R^2$ shows the coefficient of determination between GC$_{2\times2.5-200\%}$ and FLUXCOM NEE anomalies within the sub-plot.



**Figure 5.** Northern extratropical anomalies during JJA. Anomalies for (top row) $(-1) \times$SIF, (second) scPDSI, (third) $T_{soil}$, (fourth) FLUXCOM NEE, and (bottom) GC$_{2x2.5-200\%}$ NEE over JJA for (left to right) 2010–2013. Black boxes highlight major climate anomalies: the 2010 Russian heat wave, 2011 drought in Mexico and southern USA, the 2012 North American drought, and the 2013 California drought.





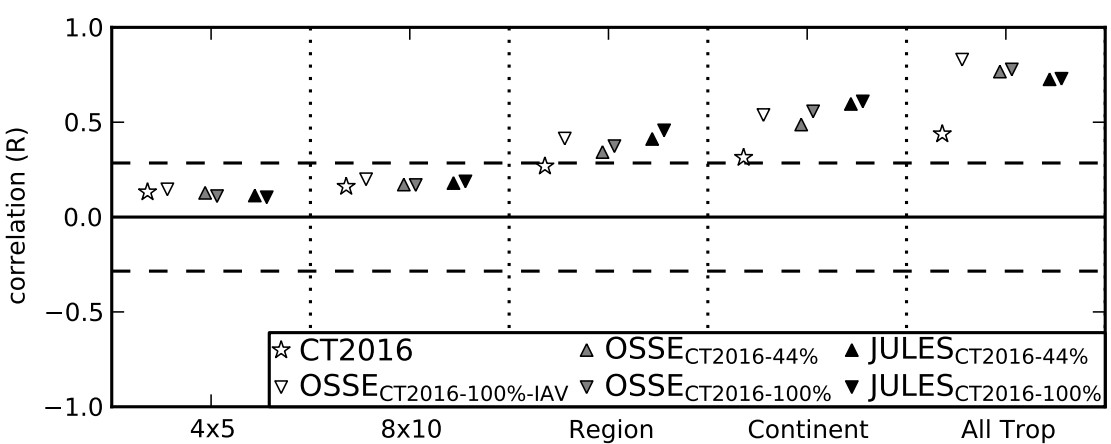

**Figure 6.** Mean correlation coefficient ($R$) with the true OSSE NEE IAV over a range of spatial scales for CT2016 NEE IAV (white star), OSSE$_{CT2016-100\%-IAV}$ NEE IAV (white down-triangle), OSSE$_{CT2016-44\%}$ NEE IAV (grey up-triangle), OSSE$_{CT2016-100\%}$ NEE IAV (grey down-triangle), OSSE$_{JULES-44\%}$ NEE IAV (black up-triangle), and OSSE$_{JULES-100\%}$ (black down-triangle) NEE IAV.



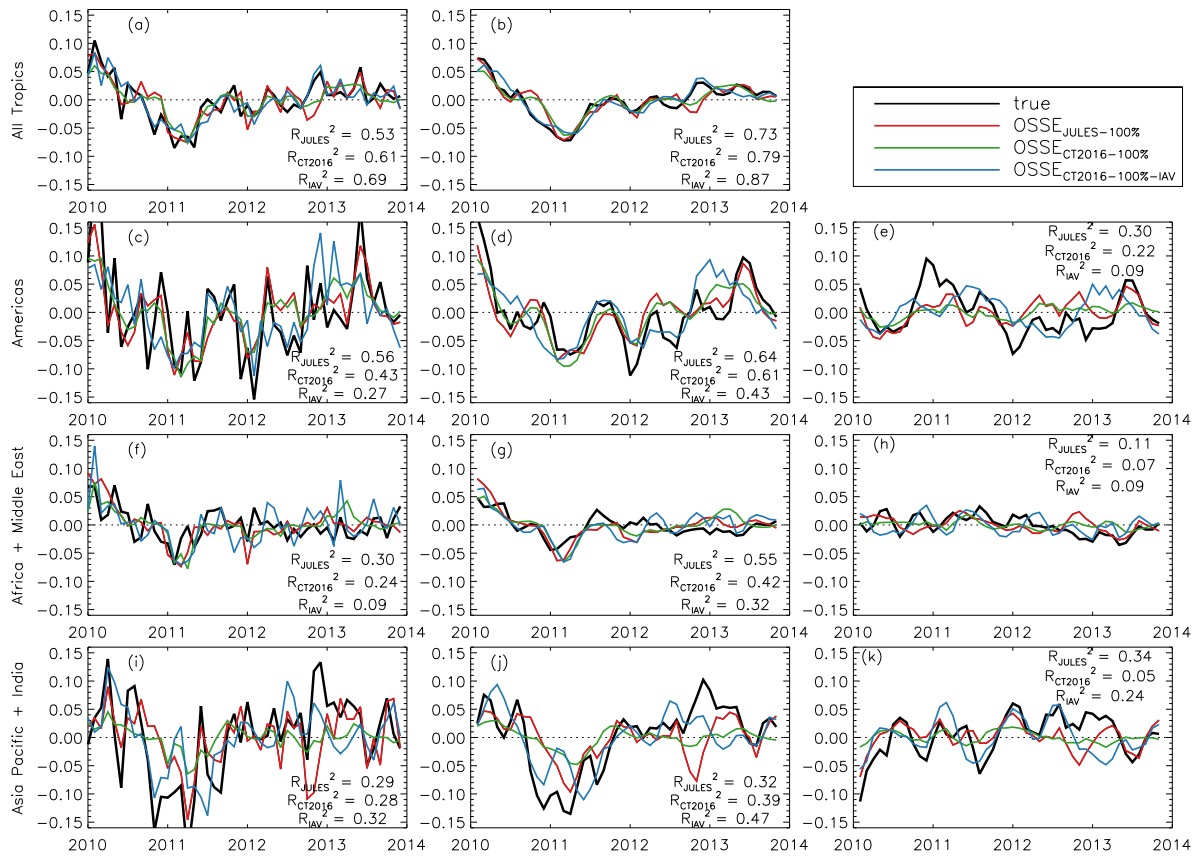

**Figure 7.** Monthly NEE anomalies (gC m$^{-2}$ day$^{-1}$) for OSSE$_{\text{JULES}-100\%}$ (red), OSSE$_{\text{CT2016}-100\%}$ (green), OSSE$_{\text{CT2016}-100\%-\text{IAV}}$ (blue) and true NEE IAV (black) in the tropics. (left column) Monthly anomalies, (center column) smoothed (3-month running mean) monthly anomalies, and (right column) DIFF$_{\text{continent}-\text{tropics}}$ (see Sect. 3.1.1 to see how this is calculated) for (a–b) the entire tropics, (c–e) the Americas, (f–h) Africa and the Middle East, and (i–k) the Asia Pacific and Indian sub-continent.



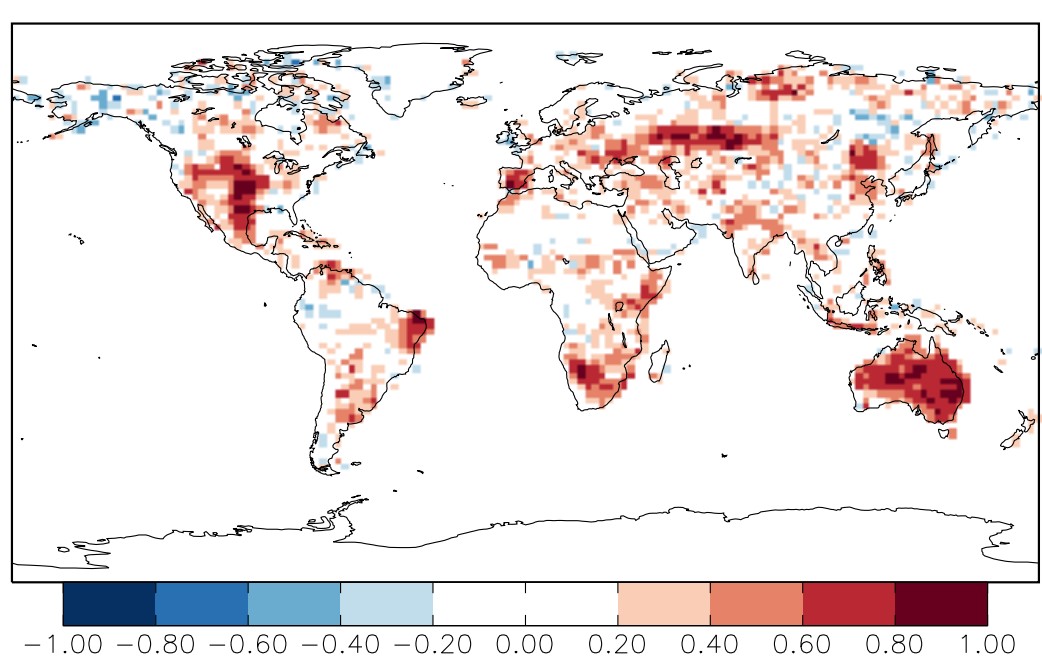

**Figure 8.** Correlation between FLUXCOM MARS GPP anomalies and SIF anomalies at $2° \times 2.5°$ spatial resolution.





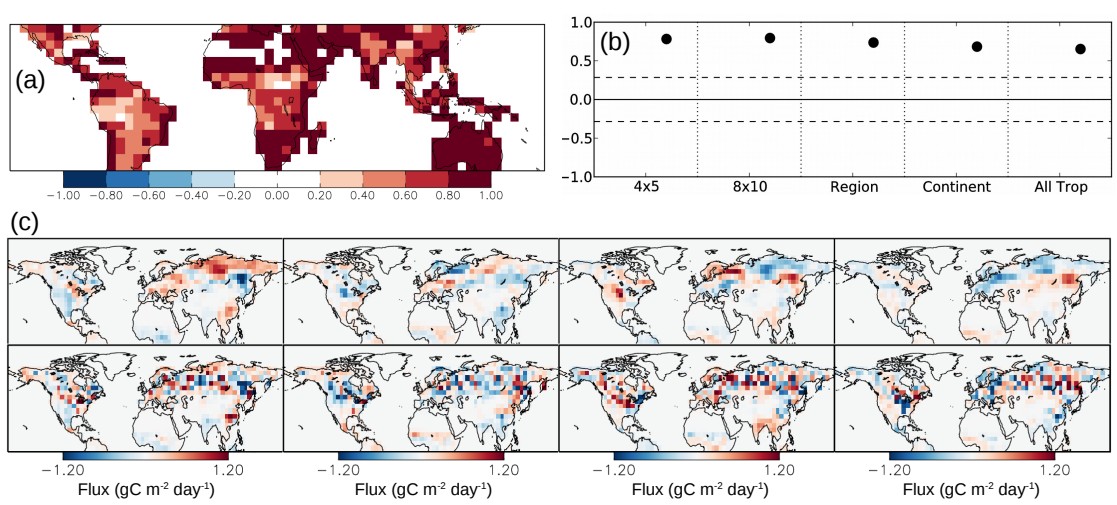

**Figure 9.** Comparison of $GC_{IAV}$ posterior and prior NEE IAV. (a) Correlation coefficient ($R$) between the posterior and prior NEE IAV in the tropics at the spatial scale of $4° × 5°$. (b) Mean correlation coefficient ($R$) between posterior and prior NEE IAV in the tropics for different degrees of spatial aggregation. (c) Northern extratropical anomalies during JJA for (top) prior and (bottom) posterior NEE for (left–right columns) 2010–2013.



**Table 1.** Setup of GEOS-Chem flux inversions. Differences are in model transport resolution, prior fluxes, and prior errors.

| Name | Resolution | Prior flux error | Prior flux IAV |
|---|---|---|---|
| $GC_{2\times2.5-200\%}$ | $2°\times2.5°$ | 200% | No (mean 2010–2013) |
| $GC_{2\times2.5-66\%}$ | $2°\times2.5°$ | 66% | No (mean 2010–2013) |
| $GC_{4\times5-100\%}$ | $4°\times5°$ | 100% | No (mean 2010–2013) |
| $GC_{4\times5-44\%}$ | $4°\times5°$ | 44% | No (mean 2010–2013) |
| $GC_{4\times5-100\%-IAV}$ | $4°\times5°$ | 100% | Yes |
| $GC_{4\times5-44\%-IAV}$ | $4°\times5°$ | 44% | Yes |





**Table 2.** Slope and coefficient of determination ($R^2$) for linear regressions of inversion/TBM NEE anomalies against proxy/FLUXCOM anomalies across the entire tropics.

| Model/Inversion $gC\,m^{-2}day^{-1}$ | FLUXCOM NEE $gC\,m^{-2}day^{-1}$ | | SIF $mW\,m^{-2}\,nm^{-1}\,sr^{-1}$ | | scPDSI | | $T_{soil}$ (K) | | NINO 3.4 index K | |
|---|---|---|---|---|---|---|---|---|---|---|
| | slope | $R^2$ | slope | $R^2$ | slope | $R^2$ | slope | $R^2$ | slope | $R^2$ |
| $GC_{2\times2.5-200\%}$ | 1.87 | 0.69 | 0.90 | 0.03 | 0.078 | 0.27 | 0.100 | 0.61 | 0.026 | 0.26 |
| $GC_{2\times2.5-66\%}$ | 1.03 | 0.62 | 0.65 | 0.05 | 0.045 | 0.27 | 0.061 | 0.66 | 0.015 | 0.26 |
| $GC_{4\times5-100\%}$ | 1.70 | 0.69 | 0.54 | 0.01 | 0.067 | 0.24 | 0.093 | 0.63 | 0.022 | 0.24 |
| $GC_{4\times5-44\%}$ | 1.06 | 0.65 | 0.65 | 0.05 | 0.044 | 0.26 | 0.061 | 0.66 | 0.014 | 0.21 |
| $GC_{4\times5-100\%-IAV}$ | 2.10 | 0.61 | 0.94 | 0.03 | 0.071 | 0.16 | 0.12 | 0.56 | 0.024 | 0.16 |
| $GC_{4\times5-44\%-IAV}$ | 1.57 | 0.51 | 0.03 | 0.00 | 0.06 | 0.16 | 0.087 | 0.55 | 0.017 | 0.12 |
| GOSAT L4 | 1.59 | 0.34 | -0.30 | 0.00 | 0.017 | 0.01 | 0.106 | 0.46 | 0.020 | 0.11 |
| GOSAT L4$_{w/BB}$ | 1.69 | 0.33 | -0.02 | 0.00 | 0.007 | 0.00 | 0.107 | 0.40 | 0.016 | 0.06 |
| CT2016 | 0.66 | 0.12 | 1.58 | 0.14 | 0.042 | 0.11 | 0.057 | 0.27 | 0.001 | 0.02 |
| CT2016$_{w/BB}$ | 0.79 | 0.14 | 1.73 | 0.14 | 0.027 | 0.04 | 0.059 | 0.24 | 0.001 | 0.00 |
| VISIT | -0.50 | 0.03 | -1.15 | 0.04 | -0.13 | 0.45 | 0.006 | 0.00 | -0.021 | 0.11 |
| CASA 4.1 | 0.38 | 0.06 | 1.88 | 0.32 | 0.030 | 0.09 | 0.023 | 0.07 | 0.004 | 0.01 |
| CASA CMS | 0.33 | 0.04 | -0.09 | 0.00 | -0.010 | 0.01 | 0.029 | 0.08 | -0.002 | 0.00 |
| JULES | 1.85 | 0.47 | 0.96 | 0.027 | 0.10 | 0.31 | 0.116 | 0.56 | 0.033 | 0.31 |



**Table 3.** Slope and coefficient of determination ($R^2$) for linear regressions of regional inversion/TBM NEE anomalies against proxy/FLUXCOM anomalies during JJA in the northern extratropics.

| Model/Inversion $gC\,m^{-2}day^{-1}$ | FLUXCOM NEE $gC\,m^{-2}day^{-1}$ | | SIF $mW\,m^{-2}\,nm^{-1}\,sr^{-1}$ | | scPDSI | | $T_{soil}$ (K) | |
|---|---|---|---|---|---|---|---|---|
| | slope | $R^2$ | slope | $R^2$ | slope | $R^2$ | slope | $R^2$ |
| $GC_{2\times2.5-200\%}$ | 1.56 | 0.54 | 4.07 | 0.14 | 0.052 | 0.21 | 0.17 | 0.56 |
| $GC_{2\times2.5-66\%}$ | 1.28 | 0.65 | 3.32 | 0.16 | 0.041 | 0.24 | 0.13 | 0.57 |
| $GC_{4\times5-100\%}$ | 1.36 | 0.49 | 4.13 | 0.17 | 0.054 | 0.28 | 0.16 | 0.62 |
| $GC_{4\times5-44\%}$ | 1.29 | 0.64 | 3.36 | 0.17 | 0.045 | 0.29 | 0.14 | 0.65 |
| $GC_{4\times5-100\%-IAV}$ | 1.28 | 0.26 | 6.8 | 0.27 | 0.05 | 0.16 | 0.16 | 0.36 |
| $GC_{4\times5-44\%-IAV}$ | 0.79 | 0.15 | 4.66 | 0.20 | 0.026 | 0.06 | 0.10 | 0.21 |
| GOSAT L4 | 1.59 | 0.33 | 5.86 | 0.17 | 0.086 | 0.35 | 0.19 | 0.43 |
| GOSAT L4$_{w/BB}$ | 1.59 | 0.34 | 6.52 | 0.21 | 0.090 | 0.39 | 0.18 | 0.39 |
| CT2016 | 0.21 | 0.01 | 4.03 | 0.13 | 0.000 | 0.00 | 0.04 | 0.03 |
| CT2016$_{w/BB}$ | 0.18 | 0.006 | 4.59 | 0.16 | 0.002 | 0.00 | 0.03 | 0.01 |
| VISIT | 0.93 | 0.47 | 3.25 | 0.21 | 0.059 | 0.67 | 0.10 | 0.50 |
| CASA 4.1 | 0.37 | 0.12 | 3.96 | 0.48 | 0.020 | 0.11 | 0.05 | 0.20 |
| CASA CMS | 0.16 | 0.01 | 4.13 | 0.34 | 0.00 | 0.00 | 0.02 | 0.02 |
| JULES | 1.58 | 0.29 | 7.26 | 0.23 | 0.075 | 0.23 | 0.23 | 0.52 |





**Table 4.** Slope and coefficient of determination ($R^2$) for linear regressions of OSSE posterior NEE anomalies against the true NEE IAV and OSSE$_{\text{JULES}-100\%}$.

| | Tropics | | | |
|---|---|---|---|---|
| Inversion | true NEE IAV | | OSSE$_{\text{JULES}-100\%}$ | |
| | slope | $R^2$ | slope | $R^2$ |
| OSSE$_{\text{JULES}-100\%}$ | 0.67 | 0.53 | | |
| OSSE$_{\text{JULES}-44\%}$ | 0.58 | 0.53 | 0.91 | 0.91 |
| OSSE$_{\text{CT2016}-100\%}$ | 0.55 | 0.61 | 0.84 | 0.84 |
| OSSE$_{\text{CT2016}-44\%}$ | 0.42 | 0.59 | 0.69 | 0.77 |
| OSSE$_{\text{CT2016}-100\%-\text{IAV}}$ | 0.75 | 0.69 | 0.70 | 0.48 |
| CT2016 | 0.31 | 0.19 | 0.50 | 0.15 |
| | Northern Extratropics | | | |
| Inversion | true NEE IAV | | OSSE$_{\text{JULES}-100\%}$ | |
| | slope | $R^2$ | slope | $R^2$ |
| OSSE$_{\text{JULES}-100\%}$ | 0.35 | 0.39 | | |
| OSSE$_{\text{JULES}-44\%}$ | 0.27 | 0.48 | 0.76 | 0.80 |
| OSSE$_{\text{CT2016}-100\%}$ | 0.30 | 0.30 | 1.04 | 0.88 |
| OSSE$_{\text{CT2016}-44\%}$ | 0.31 | 0.43 | 1.06 | 0.62 |
| OSSE$_{\text{CT2016}-100\%-\text{IAV}}$ | 0.63 | 0.15 | 0.55 | 0.41 |
| CT2016 | 0.48 | 0.46 | 0.18 | 0.05 |