# Peer review of "On what scales can GOSAT flux inversions constrain anomalies in terrestrial ecosystems?"

_Atmospheric Chemistry and Physics, 2018_

## Referee Comment (RC1) · Anonymous Referee #1 · 22 Jan 2019

———————— Review of Byrne et al. ————————

**Holistic suggestions**

The authors have written an interesting paper that is very relevant to current inverse modeling efforts using satellite data.

Overall, I think the science in this paper is very sound. The authors have been very thorough and thoughtful. With that said, the manuscript contains enormous detail and discussion. I worry that the main takeaway messages can get lost in this detail. I think the main science question posed in this paper could have a relatively straightforward answer; I think one could answer that question in a more succinct way that brings to the forefront the unique results of the paper. Here are a few ways that one could shorten

the paper: (1) Some text in the introduction seems better-suited for the methods (see below), and I think the intro could end with a more focused, succinct list of the study goals. (2) Some of the details on the proxies and the flux products could go into a Supplement, though I think it could go either way. (3) I would consider merging the results and discussion into a single section. These two sections can feel repetitive in places. (4) I think the Conclusions could be half the current length and focus on bringing the analysis together into snappy, punchy take-away messages. These are just a few ideas, and I think the authors should make whatever changes feel most appropriate to them. I want to be sure this article gets the attention it deserves when published and ensure that the authors don't lose readers in the detail.

Below are several detailed suggestions.

**Abstract**

- You may want to either define FLUXCOM or use a non-technical term here.

- What is $R_{NINO3.4}$? Again, you might want to either define this term or use a less technical term.

- Line 14: Could you define "regional scales" more precisely? I.e., what size regions are you referring to here?

- Line 14: It could be helpful to give the reader a hint on what aspects of the configuration you're referring to here.

**Introduction and methods**

- pg. 2, Line 23, "We also examine the posterior fluxes from two publicly available flux inversion estimates....": Can you provide a bit more detail here. In what way do you examine these fluxes, and how does this examination contribute to the study?

- I would write out abbreviations like MERRA-2, FLUXNET, and FLUXCOM the first time you use them.

- Pg. 2, line 32 to pg. 4, line 9: The introduction is relatively long. You could condense or eliminate some of this information to produce a shorter, punchier introduction.

- Pg. 4, line 3 to line 15: This material could be a better fit for the methods sections.

- Pg. 4, lines 16-27: Again, I think you could shorten the descriptions here to make the make the main objectives more concise and punchy. There's a lot of detail to digest in this paragraph. It could be helpful to give a more succinct overview of the paper and leave some of the details for later.

- Pg. 7, line 3: I recommend experimenting with covariance matrices that include off-diagonal elements. I worry that you could overestimate the information content of the satellite observations by using a diagonal observational covariance matrix. Existing studies suggest that GOSAT retrievals have spatially and temporally coherent errors. My guess is that the estimated fluxes would be less detailed or less informative if you included off-diagonal elements in the observational covariance matrix. The overall goal of this study is to estimate the robustness of the flux constraint using GOSAT observations, and I think that the structure of the covariance matrices could have an important impact on these conclusions.

**Results and conclusions**

- Pg. 10, line 26, "As scales decrease....": It could be clearer to rephrase to "As the size of the regions decrease....".

- Pg. 10, line 32: I would be clearer about what you mean by "regional and larger scales." Different readers might have different concepts of what this phrase means.

Interactive
comment

[Figure]

- In general, there's a lot of detail in Sect. 3. I think you could shorten this section and/or potentially move some of the text to an SI without losing key information. An upside of being so thorough is that you won't omit anything from the text. However, a danger is that the key messages of the paper can become buried in all of the detailed descriptions.

- Sect. 3.1.1 title: I think it would be more appealing to a reader to use a descriptive title here instead of using a relatively technical abbreviation in the header title. Someone who is scanning through the paper wouldn't know what $GC_{2x2.5-200\%}$ is or what the scientific importance of this section is.

- Pg. 14, line 18: Consider starting this section with a stronger topic sentence that helps guide the reader through the material. For example, you could state the main take-away message of the OSSEs in the tropics and then explain the results in subsequent sentences.

- Sect. 3.3.1: I think you could add a few more sentences somewhere in this section explaining what the implications of the OSSE's are for your real data results. What have you learned from the OSSE's, and what are the implications for using GOSAT to constrain flux anomalies?

- Pg. 20, line 1: I think it is a bit too informal to start a paragraph with a question. You might want to rephrase with a guiding topic sentence (e.g., one that states the main result or take-away of the paragraph).

- You use the term "flux anomalies" a lot throughout the paper. I would make sure it's very clear to the reader what type of anomalies you are referring to (e.g., over what time periods). I know that you specify this information in the Methods, but it doesn't hurt to use very clear and unambiguous terminology whenever possible.

- Pg. 21, line 19 "significant impact": Is this result statistically significant? I would

reserve the word "significant" for instances when you explicitly tested for statistical significance. Otherwise, you could unintentionally confuse a reader.

---

## Referee Comment (RC2) · Abhishek Chatterjee (Referee) · 2 Aug 2019

This is a highly relevant and high-quality study that makes use of the ever-increasing time record of satellite retrievals (in this case JAXA's GOSAT), and demonstrates their value in providing the community with new insights into carbon cycle science. While most inversion studies tend to diagnose carbon fluxes, this study makes a valuable attempt at attribution by relating the carbon flux anomalies with various auxiliary environmental variables (called proxies here). The manuscript is well-written and the figures and text are of high-quality. Before recommending final publication though, I would highly advise the authors and the ACP editorial team to make note of the following comments and suggestions:

[Figure]

Major Comments:

(1) In many ways, the manuscript reads like a dissertation chapter. There are too many details that are unnecessary for a manuscript as it outlines all the sensitivity tests and checks that were done. I would highly recommend that the authors think about what the main takeaway messages are, and accordingly cull the text from Sections 3, 4 and 5 to focus on those messages. There are some strong points made in the paper, for example - the need to use prior flux estimates that have a realistic seasonal cycle and IAV. But more often than not, these types of important messages are lost in the text. Even though the manuscript is well-written, it is cumbersome to read. Some of the sensitivity tests could be moved to Supplementary Information. Also, see comments #2 & #4 below on which text are most suited for this.

(2) Is SIF really needed? As the authors rightly point out in multiple places (Page 3, Lines 24-29, Page 11, discussion in Lines 17-30, Page 16, Lines 30-32), SIF is a good proxy for GPP. Its relationship to NEE is complicated by the role of R. If the authors were estimating GPP and R separately, then SIF would have been a valuable dataset to correlate against. But given that the authors estimate NEE, it is impractical to assume SIF will provide any new information or help constrain the scales at which GOSAT can inform NEE fluxes. Results presented in Figure 2 and Table 2 further show the redundancy of SIF for this paper. In fact, it is not surprising at all that SIF is well correlated with CASA GFED ( a model based on light-use efficiency) and consequently CT-2016 (that uses CASA-GFED as its prior). Finally, Section 4.1.2 doesn't add any new information to the study at all. By taking out SIF, the authors could shorten the paper and make it more focused.

(3) Follow-on to the previous comment - if the authors were interested in using a vegetation index, they may want to look at something more direct like LAI. Or even indices like NDVI, EVI, fPAR x PAR may act as robust proxy capturing the interaction between vegetation and radiation.

(4) Should the OSSEs be moved to the Supplementary Material? While the scientific rationale for doing the OSSEs make sense, they add a new set of results that are frankly not necessary. The real data inversions already cover all the conclusions from the OSSEs. I understand that the OSSEs may have been done initially to ensure that the GEOS-Chem 4DVAR system works, before the authors move to a real-data inversion. But given the more interesting and thought provoking results with the real data, Section 3.3 seems superfluous.

(5) The attempt to isolate anomalies specific to each continent (Equation 2) is unclear. I do not follow how the standard deviation of the tropical and the continental anomaly are calculated in the first place. Do the authors do some type of Monte Carlo simulation (i.e., an ensemble of fluxes and anomalies) to calculate STD(ANOM).

(6) A critical component that is missing from the study is the calculation of time-lagged correlations. The impact of ENSO events or droughts are not immediate (i.e., within a month) on fluxes. There is a spatiotemporal cascade of this impact across different tropical and Northern extratropical regions. In fact the authors do acknowledge this briefly (Page 16, Lines 17-20). Also see Rayner et al. 1999 (GRL, Vol. 26(4)). My hypothesis is that both the R2 and the significance of several of the flux estimates will change when the authors use time-lagged correlation, especially with the Nino3.4 index. Can the authors comment further on why they didn't pursue? A figure or two in Supplementary Materials may help. (6) Finally, why isn't there a run at 2 x 2.5 with prescribed IAV in the prior flux? If anything, I would want to see how an inversion run at 2 x 2.5 with 200% uncertainty applied to prior fluxes and with prior NEE IAV performs. The results may really enforce the message in Section 4.3.3 about the realism of IAV in prior fluxes and the important role it plays on the posterior IAV that is recovered during the inversion.

Minor Comments:

(1) Page 5, Section 2.1 - What is the spatial and temporal resolution at which FLUX-

COM products are available? Are there potentially issues with scale mismatch that the reader should be aware of?

(2) Page 7, Lines 30-33 -The current nomenclature for the different runs is extremely complex. I do not know if there is a way of making the names more intuitive and/or easy to follow. But something the authors can think about a bit more.

(3) Page 8, Line 3 - Did GOSAT ACOS v3.5 had warn levels? Or just quality flags? To the best of my knowledge warn levels weren't implemented till v7.3. Kindly check.

(4) Page 9, Equation 1 - It is worth making the reader aware of the drawbacks of using a short period to define the climatology.

(5) Section 4 Discussion topics - what about the discussion for agreement between flux anomalies and Nino 3.4 index? That discussion seems to have gone missing.

- Abhishek Chatterjee, Global Modeling and Assimilation Office (GMAO), NASA Goddard Space Flight Center, Email: abhishek.chatterjee@nasa.gov, Url: http://gmao.gsfc.nasa.gov/bio600/abhishek.chatterjee.html

---

## Author Comment (AC1) · 13 Sep 2019

We thank Abhishek Chatterjee and an anonymous reviewer for their constructive comments. We have addressed the comments of the reviewers below. Reviewer comments are shown in bold and changes to the manuscript are underlined. Line numbers refer to the marked changes manuscript.

Both reviewers found that the manuscript contained excessive detail that obscured the main results of the study. We have gone through the manuscript to remove excessive detail and present the results more concisely. In particular, we have shortened the introduction, removed any discussion from the results section, and shortened the discussion section. Some of the analysis that is complementary to the main results has been moved to the supplementary analysis. These steps have reduced the length of the manuscript by about 10 pages.

Response to reviewer 1:

**Holistic suggestions**

**The authors have written an interesting paper that is very relevant to current inverse modeling efforts using satellite data. Overall, I think the science in this paper is very sound. The authors have been very thorough and thoughtful. With that said, the manuscript contains enormous detail and discussion. I worry that the main takeaway messages can get lost in this detail. I think the main science question posed in this paper could have a relatively straightforward answer; I think one could answer that question in a more succinct way that brings to the forefront the unique results of the paper. Here are a few ways that one could shorten the paper: (1) Some text in the introduction seems better-suited for the methods (see below), and I think the intro could end with a more focused, succinct list of the study goals. (2) Some of the details on the proxies and the flux products could go into a Supplement, though I think it could go either way. (3) I would consider merging the results and discussion into a single section. These two sections can feel repetitive in places. (4) I think the Conclusions could be half the current length and focus on bringing the analysis together into snappy, punchy take-away messages. These are just a few ideas, and I think the authors should make whatever changes feel most appropriate to them. I want to be sure this article gets the attention it deserves when published and ensure that the authors don't lose readers in the detail. Below are several detailed suggestions.**

We have gone through the manuscript and tried to remove any unnecessary text and reorganized some of the text to improve the clarity of the manuscript.

**Abstract**

**• You may want to either define FLUXCOM or use a non-technical term here.**

As far as we are aware, FLUXCOM is not an acronym but the name of a flux product. We have added the URL for the datasets immediately after the first occurrence of "FLUXCOM".

**• What is RNINO3.4? Again, you might want to either define this term or use a less technical term.**

In the abstract, we have changed the text to state the "statistical significance" (P<0.05) rather than using this term, as the reader will likely be familiar with this term.

**• Line 14: Could you define "regional scales" more precisely? I.e., what size regions are you referring to here?**

Clarified to "large sub-continental regions"

**• Line 14: It could be helpful to give the reader a hint on what aspects of the configuration you're referring to here.**

This statement was removed from the abstract.

**Introduction and methods**

**• pg. 2, Line 23, "We also examine the posterior fluxes from two publicly available flux inversion estimates....": Can you provide a bit more detail here. In what way do you examine these fluxes, and how does this examination contribute to the study?**

This sentence has been re-worded:

P3L1-2: "NEE anomalies produced by the GEOS-Chem flux inversions are contrasted with two independent publicly available flux inversion estimates."

**• I would write out abbreviations like MERRA-2, FLUXNET, and FLUXCOM the first time you use them.**

Done for MERRA2. FLUXNET and FLUXCOM are not acronyms, as far as we are aware. We have added the URL for the datasets immediately after the first occurrence of FLUXNET and FLUXCOM.

**• Pg. 2, line 32 to pg. 4, line 9: The introduction is relatively long. You could condense or eliminate some of this information to produce a shorter, punchier introduction.**

We have tried to condense the introduction, and have shortened it.

**• Pg. 4, line 3 to line 15: This material could be a better fit for the methods sections.**

We have removed this material from the introduction.

**• Pg. 4, lines 16-27: Again, I think you could shorten the descriptions here to make the make the main objectives more concise and punchy. There's a lot of detail to digest in this paragraph. It could be helpful to give a more succinct overview of the paper and leave some of the details for later.**

We have tried to do this; the introduction has been restructured for this purpose.

**• Pg. 7, line 3: I recommend experimenting with covariance matrices that include off-diagonal elements. I worry that you could overestimate the information content of the satellite observations by using a diagonal observational covariance matrix. Existing studies suggest that GOSAT retrievals have spatially and temporally coherent errors. My guess is that the estimated fluxes would be less detailed or less informative if you included off-diagonal elements in the observational covariance matrix. The overall goal of this study is to estimate the robustness of the flux constraint using GOSAT observations, and I think that the structure of the covariance matrices could have an important impact on these conclusions.**

We agree that there are likely spatially and temporally coherent errors in the ACOS GOSAT measurements. However, implementing off-diagonal elements in our inversion set-up is not straight forward, and would be a study on its own. Thus, this is beyond the scope of this study. We have added a statement to the conclusions encouraging future research in this field:

P24L20-22: "Although not addressed in this study, correlated errors between GOSAT observations may introduce structures in the posterior NEE estimates, thus we recommend future work address the possibility of prescribing non-diagonal in the observational error covariance matrix."

**Results and conclusions**

**• Pg. 10, line 26, "As scales decrease....": It could be clearer to rephrase to "As the size of the regions decrease....".**

Changed to "At smaller spatial scales"

**• Pg. 10, line 32: I would be clearer about what you mean by "regional and larger scales." Different readers might have different concepts of what this phrase means.**

We have specified "sub-continental regions" whenever we mention "regional" scales

**• In general, there's a lot of detail in Sect. 3. I think you could shorten this section and/or potentially move some of the text to an SI without losing key information. An upside of being so thorough is that you won't omit anything from the text. However, a danger is that the key messages of the paper can become buried in all of the detailed descriptions.**

We have removed a substantial amount of text from Section 3, and ensured that the remaining text only described the results without giving a discussion.

**• Sect. 3.1.1 title: I think it would be more appealing to a reader to use a descriptive title here instead of using a relatively technical abbreviation in the header title. Someone who is scanning through the paper wouldn't know what GC$_{2x2.5-200\%}$ is or what the scientific importance of this section is.**

This section has been moved to supplementary materials and renamed to "Detailed analysis of tropical NEE anomalies and ENSO".

**• Pg. 14, line 18: Consider starting this section with a stronger topic sentence that helps guide the reader through the material. For example, you could state the main take-away message of the OSSEs in the tropics and then explain the results in subsequent sentences.**

Changed topic sentence:

P15L18: "Strong correlations are obtained between the posterior and true anomalies for all OSSEs on sub-continental regional and larger scales, suggesting that sub-continental regions are the minimum scales that can be constrained by GOSAT measurements."

**• Sect. 3.3.1: I think you could add a few more sentences somewhere in this section explaining what the implications of the OSSE's are for your real data results. What have you learned from the OSSE's, and what are the implications for using GOSAT to constrain flux anomalies?**

Added to beginning of the section:

P15L10:" Strong correlations between the GOSAT flux inversions and proxies/FLUXCOM provide evidence that the GOSAT flux inversions give realistic constraints on NEE. However, the absence of strong correlations does not imply that the GOSAT flux inversions are not constraining IAV as there could be other causes (such as lagged effects within ecosystems) that can explain the absence of correlations. Therefore, to investigate the minimum spatial scales that can be constrained by GOSAT observations, we performed a series of OSSE experiments. In these experiments pseudo-observations were assimilated from a GEOS-Chem forward model run which had JULES NEE fluxes prescribed."

**• Pg. 20, line 1: I think it is a bit too informal to start a paragraph with a question. You might want to rephrase with a guiding topic sentence (e.g., one that states the main result or take-away of the paragraph).**

Removed this sentence, and tried to make this section more concise.

**• You use the term "flux anomalies" a lot throughout the paper. I would make sure it's very clear to the reader what type of anomalies you are referring to (e.g., over what time periods). I know that you specify this information in the Methods, but it doesn't hurt to use very clear and unambiguous terminology whenever possible.**

We have edited the manuscript throughout to clarify quantity and time interval.

**• Pg. 21, line 19 "significant impact": Is this result statistically significant? I would reserve the word "significant" for instances when you explicitly tested for statistical significance. Otherwise, you could unintentionally confuse a reader.**

Removed.

Response to Reviewer 2:

**This is a highly relevant and high-quality study that makes use of the ever-increasing time record of satellite retrievals (in this case JAXA's GOSAT), and demonstrates their value in providing the community with new insights into carbon cycle science. While most inversion studies tend to diagnose carbon fluxes, this study makes a valuable attempt at attribution by relating the carbon flux anomalies with various auxiliary environmental variables (called proxies here). The manuscript is well-written and the figures and text are of high-quality. Before recommending final publication though, I would highly advise the authors and the ACP editorial team to make note of the following comments and suggestions:**

**Major Comments:**

(1) **In many ways, the manuscript reads like a dissertation chapter. There are too many details that are unnecessary for a manuscript as it outlines all the sensitivity tests and checks that were done. I would highly recommend that the authors think about what the main takeaway messages are, and accordingly cull the text from Sections 3, 4 and 5 to focus on those messages. There are some strong points made in the paper, for example - the need to use prior flux estimates that have a realistic seasonal cycle and IAV. But more often than not, these types of important messages are lost in the text. Even though the manuscript is well-written, it is cumbersome to read. Some of the sensitivity tests could be moved to Supplementary Information. Also, see comments #2 & #4 below on which text are most suited for this.**

We have gone through the manuscript and tried to streamline to highlight the main points. The manuscript is now substantially shorter.

(2) **Is SIF really needed? As the authors rightly point out in multiple places (Page 3, Lines 24-29, Page 11, discussion in Lines 17-30, Page 16, Lines 30-32), SIF is a good proxy for GPP. Its relationship to NEE is complicated by the role of R. If the authors were estimating GPP and R separately, then SIF would have been a valuable dataset to correlate against. But given that the authors estimate NEE, it is impractical to assume SIF will provide any new information or help constrain the scales at which GOSAT can inform NEE fluxes. Results presented in Figure 2 and Table 2 further show the redundancy of SIF for this paper. In fact, it is not surprising at all that SIF is well correlated with CASA GFED ( a model based on light-use efficiency) and consequently CT-2016 (that uses CASA-GFED as its prior). Finally, Section 4.1.2 doesn't add any new information to the study at all. By taking out SIF, the authors could shorten the paper and make it more focused.**

We prefer to retain SIF in the manuscript, as this is the only proxy that is a direct measure of ecosystem functioning (even if it is a measure of GPP). Furthermore, we do find substantial correlations between SIF anomalies and NEE anomalies in the Northern extra

tropics. Similarly, we see that large anomalies and heatwaves are shown to have a large impact on ecosystem functioning in Figure 3.

**(3) Follow-on to the previous comment - if the authors were interested in using a vegetation index, they may want to look at something more direct like LAI. Or even indices like NDVI, EVI, fPAR x PAR may act as robust proxy capturing the interaction between vegetation and radiation.**

There are many interesting potential "proxies" for ecosystem functioning that we could examine. However, we think that the inclusion of additional proxies would distract from our main point that we can see strong correlations with the proxies chosen for this study on a variety of scales.

**(4) Should the OSSEs be moved to the Supplementary Material? While the scientific rationale for doing the OSSEs make sense, they add a new set of results that are frankly not necessary. The real data inversions already cover all the conclusions from the OSSEs. I understand that the OSSEs may have been done initially to ensure that the GEOS-Chem 4DVAR system works, before the authors move to a real-data inversion. But given the more interesting and thought provoking results with the real data, Section 3.3 seems superfluous.**

We prefer to retain the OSSEs in the main text. This is because the OSSEs are required to estimate the minimum scales constrained by the GOSAT measurements and estimate the magnitude of posterior IAV relative to true IAV. Neither of these aspects of the GOSAT flux inversions can be examined in the real data experiments.

**(5) The attempt to isolate anomalies specific to each continent (Equation 2) is unclear. I do not follow how the standard deviation of the tropical and the continental anomaly are calculated in the first place. Do the authors do some type of Monte Carlo simulation (i.e., an ensemble of fluxes and anomalies) to calculate STD(ANOM).**

It is calculated as the standard deviation of the monthly anomalies. We have decided to move this discussion to the supplementary materials as the main conclusions of the analysis could be made without this analysis. We have the revised the sentence in the supplementary materials to state "standard deviation of monthly anomalies".

**(6) A critical component that is missing from the study is the calculation of time-lagged correlations. The impact of ENSO events or droughts are not immediate (i.e., within a month) on fluxes. There is a spatiotemporal cascade of this impact across different tropical and Northern extratropical regions. In fact the authors do acknowledge this briefly (Page 16, Lines 17-20). Also see Rayner et al. 1999 (GRL, Vol. 26(4)). My hypothesis is that both the R2 and the significance of several of the flux estimates will change when the authors use time-lagged correlation, especially with the Nino3.4 index. Can the authors comment further on why they didn't pursue? A figure or two in Supplementary Materials may help.**

We agree that there are substantial lagged effects between climate anomalies and NEE anomalies, and that these lagged effects are important for understanding the carbon cycle. However, we have intentionally avoided discussing lagged effects in this manuscript to simplify the analysis. As pointed out by the reviewer, we do acknowledge that these lagged effects exist in the manuscript. We hope that these effects will be investigated further in future studies, and have suggested this is the revised conclusions:

P24L26-31:"The results of this study suggest that GOSAT measurements provide a useful constraint on IAV in the carbon cycle. Further study of the relationship between GOSAT-constrained NEE and environmental variables is merited given results discussed here. In particular, the mechanisms driving these co-variations should be further investigated. Lagged relationships between GOSAT-constrained NEE and environmental variables should also be investigated. Future research could also investigate differences in IAV between GOSAT-constrained NEE and that produced by TBMs. Given the better agreement with the proxies, GOSAT-constrained NEE IAV may provide a tool for evaluating the TBM-simulated NEE IAV in the future."

**(7) Finally, why isn't there a run at 2 x 2.5 with prescribed IAV in the prior flux? If anything, I would want to see how an inversion run at 2 x 2.5 with 200% uncertainty applied to prior fluxes and with prior NEE IAV performs. The results may really enforce the message in Section 4.3.3 about the realism of IAV in prior fluxes and the important role it plays on the posterior IAV that is recovered during the inversion.**

This was not performed because of the computation requirements. Performing 4DVar at 2x2.5 spatial resolution with the GEOS-Chem is extremely computationally expensive, thus we tried to perform all of the sensitivity tests at 4x5 degrees.

**Minor Comments:**

**(1) Page 5, Section 2.1 - What is the spatial and temporal resolution at which FLUXCOM products are available? Are there potentially issues with scale mismatch that the reader should be aware of?**

The FLUXCOM products are generated at 0.5x0.5 degrees resolution. This has been added to the text.

P5L18: "FLUXCOM remote sensing and meteorological data (RS+METEO) products are generated at 0.5 x 0.5 spatial resolution using upscaling approaches based on machine learning methods that integrate FLUXNET site level observations, satellite remote sensing, and meteorological data"

**(2) Page 7, Lines 30-33 -The current nomenclature for the different runs is extremely complex. I do not know if there is a way of making the names more intuitive and/or easy to follow. But something the authors can think about a bit more.**

We tried a number of different acronyms, but found the current set-up to be the clearest. We hope that table 1 will provide a useful tool to remind the reader of the set-up.

**(3) Page 8, Line 3 - Did GOSAT ACOS v3.5 had warn levels? Or just quality flags? To the best of my knowledge warn levels weren't implemented till v7.3. Kindly check.**

Warn levels were implemented in ACOS 3.5. This was the first ACOS GOSAT product to have them.

**(4) Page 9, Equation 1 - It is worth making the reader aware of the drawbacks of using a short period to define the climatology.**

We have added the following text.

P10L9-11: "It is worth noting that four years is a relatively short period to define a climatology, and some modes of climate variability occur on longer timescales. Ideally, a longer time period would be used to calculate a climatology, but we are limited by the availability of GOSAT data in this study."

**(5) Section 4 Discussion topics - what about the discussion for agreement between flux anomalies and Nino 3.4 index? That discussion seems to have gone missing.**

This has been added as Section 4.2.